# Pin-Tuning: Parameter-Efficient In-Context Tuning for Few-Shot Molecular Property Prediction

**Liang Wang**[1 2]  **Qiang Liu**[1 2 †]  **Shaozhen Liu**[1]  **Xin Sun**[3]  **Shu Wu**[1 2]  **Liang Wang**[1 2 3]

[1]New Laboratory of Pattern Recognition (NLPR)
State Key Laboratory of Multimodal Artificial Intelligence Systems (MAIS)
Institute of Automation, Chinese Academy of Sciences (CASIA)
[2] School of Artificial Intelligence, University of Chinese Academy of Sciences
[3] University of Science and Technology of China

liang.wang@cripac.ia.ac.cn, qiang.liu@nlpr.ia.ac.cn, liushaozhen2025@ia.ac.cn
sunxin000@mail.ustc.edu.cn, {shu.wu, wangliang}@nlpr.ia.ac.cn

## Abstract

Molecular property prediction (MPP) is integral to drug discovery and material science, but often faces the challenge of data scarcity in real-world scenarios. Addressing this, few-shot molecular property prediction (FSMPP) has been developed. Unlike other few-shot tasks, FSMPP typically employs a pre-trained molecular encoder and a context-aware classifier, benefiting from molecular pre-training and molecular context information. Despite these advancements, existing methods struggle with the ineffective fine-tuning of pre-trained encoders. We attribute this issue to the imbalance between the abundance of tunable parameters and the scarcity of labeled molecules, and the lack of contextual perceptiveness in the encoders. To overcome this hurdle, we propose a parameter-efficient in-context tuning method, named `Pin-Tuning`. Specifically, we propose a lightweight adapter for pre-trained message passing layers (`MP-Adapter`) and Bayesian weight consolidation for pre-trained atom/bond embedding layers (`Emb-BWC`), to achieve parameter-efficient tuning while preventing over-fitting and catastrophic forgetting. Additionally, we enhance the `MP-Adapters` with contextual perceptiveness. This innovation allows for in-context tuning of the pre-trained encoder, thereby improving its adaptability for specific FSMPP tasks. When evaluated on public datasets, our method demonstrates superior tuning with fewer trainable parameters, improving few-shot predictive performance.[‡]

## 1 Introduction

In the field of drug discovery and material science, molecular property prediction (MPP) stands as a pivotal task [5, 9, 63]. MPP involves the prediction of molecular properties like solubility and toxicity, based on their structural and physicochemical characteristics, which is integral to the development of new pharmaceuticals and materials. However, a major challenge encountered in real-world MPP scenarios is data scarcity. Obtaining extensive molecular data with well-characterized properties can be time-consuming and expensive. To address this, few-shot molecular property prediction (FSMPP) has emerged as a crucial approach, enabling predictions with limited labeled molecules [1, 41, 4].

The methodology for general MPP typically adheres to an encoder-classifier framework [71, 23, 27, 56], as illustrated in Figure 2(a). In this streamlined framework, the encoder converts molecular

---

[†]Corresponding author
[‡]Code is available at: https://github.com/CRIPAC-DIG/Pin-Tuning

38th Conference on Neural Information Processing Systems (NeurIPS 2024).

structures into vectorized representations [12, 28, 50, 67, 2], and then the classifier uses these representations to predict molecular properties. In the context of few-shot scenarios, two significant discoveries have been instrumental in advancing this task. Firstly, *pre-trained molecular encoders* have demonstrated consistent effectiveness in FSMPP tasks [20, 14, 58]. This indicates the utility of leveraging pre-acquired knowledge in dealing with data-limited scenarios. Secondly, unlike typical few-shot tasks such as image classification [57, 48], FSMPP tasks greatly benefits from *molecular context information*. This involves comprehending the seen many-to-many relationships between molecules and properties [58, 45, 73], as molecules are multi-labeled by various properties. These two discoveries have collectively led to the development of the widely used FSMPP framework that utilizes a pre-trained encoder followed by a context-aware classifier, as shown in Figure 2(b).

Despite the progress, there are observed limitations in the current approaches to FSMPP. Notably, while using a pre-trained molecular encoder generally outperforms training from scratch, fine-tuning the pre-trained encoder often leads to inferior results compared to keeping it frozen, which can be observed in Figure 1.

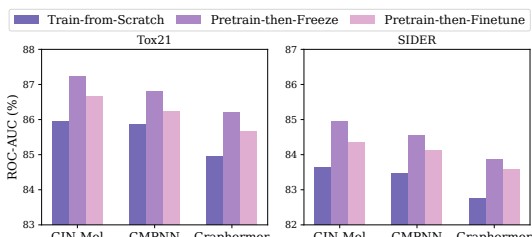

The observed ineffective fine-tuning can be attributed to two primary factors: (i) Imbalance between the abundance of tunable parameters and the scarcity of labeled molecules: fine-tuning all parameters of a pre-trained encoder with few labeled molecules leads to a disproportionate ratio of tunable parameters to available data. This imbalance often results in over-fitting and catastrophic forgetting [7, 6]. (ii) Limited contextual perceptiveness in the encoder: while molecular

Figure 1: Comparison of molecular encoders trained via different paradigms: train-from-scratch, pretrain-then-freeze, and pretrain-then-finetune. The evaluation is conducted across two datasets and three encoder architectures [20, 47, 66]. The results consistently demonstrate that while pretraining outperforms training from scratch, the current methods do not yet effectively facilitate finetuning.

context is leveraged to enhance the classifier [58, 73], the encoder typically lacks the explicit capability to perceive this context, relying instead on implicit gradient-based optimization. This leads to the encoder not directly engaging with the nuanced molecular context information that is critical in FSMPP tasks. In summary, while significant strides have been made, the challenges of imbalance between the number of parameters and labeled data, along with the need for contextual perceptiveness in the encoder, necessitate more sophisticated methodologies in this domain.

Based on the aforementioned analysis, we propose the parameter-efficient in-context tuning method, named `Pin-Tuning`, to address the two primary challenges in FSMPP. To overcome the parameter-data imbalance, we propose a parameter-efficient chemical knowledge adaptation approach for pre-trained molecular encoders. A lightweight adapters (`MP-Adapter`) are designed to tune the pre-trained message passing layers efficiently. Additionally, we impose a Bayesian weight consolidation (`Emb-BWC`) on the pre-trained embedding layers to prevent aggressive parameter updates, thereby mitigating the risk of over-fitting and catastrophic forgetting. To address the second challenge, we further endow the `MP-Adapter` with the capability to perceive context. This innovation allows for in-context tuning of the pre-trained molecular encoders, enabling them to adapt more effectively to specific downstream tasks. Our approach is rigorously evaluated on public datasets. The experimental results demonstrate that our method achieves superior tuning performance with fewer trainable parameters, leading to enhanced performance in few-shot molecular property prediction.

The main contributions of our work are summarized as follows:

- We analyze the deficiencies of existing FSMPP approaches regarding the adaptation of pre-trained molecular encoders. The key issues include an imbalance between the number of tunable parameters and labeled molecules, as well as a lack of contextual perceptiveness in the encoders.

- We propose `Pin-Tuning` to adapt the pre-trained molecular encoders for FSMPP tasks. This includes the `MP-Adapter` for message passing layers and the `Emb-BWC` for embedding layers, facilitating parameter-efficient tuning of pre-trained molecular encoders.

- We further endow the `MP-Adapter` with the capability to perceive context to allows for in-context tuning, which provides more meaningful adaptation guidance during the tuning process.

- We conduct extensive experiments on benchmark datasets, which show that `Pin-Tuning` outperforms state-of-the-art methods on FSMPP by effectively tuning pre-trained molecular encoders.

## 2   Related work

**Few-shot molecular property prediction.** Few-shot molecular property prediction aims to accurately predict the properties of new molecules with limited training data [49]. Early research applied general few-shot techniques to FSMPP. IterRefLSTM [1] is the pioneer work to leverage metric learning to solve FSMPP problem. Following this, Meta-GGNN [41] and Meta-MGNN [14] introduce meta-learning with graph neural networks, setting a foundational framework that subsequent studies have continued to build upon [39, 40, 4]. It is noteworthy that Meta-MGNN employs a *pre-trained molecular encoder* [20] and achieves superior results through fine-tuning in the meta-learning process compared to training from scratch. In fact, pre-trained graph neural networks [64, 36, 17, 54, 37] have shown promise in enhancing various graph-based downstream tasks [52, 13], including molecular property prediction [60, 62, 38, 72]. Recent efforts have shifted towards leveraging unique nature in FSMPP, such as the many-to-many relationships between molecules and properties arising from the multi-labeled nature of molecules, often referred to as the *molecular context*. PAR [58] initially employs graph structure learning [32, 55] to connect similar molecules through a homogeneous context graph. MHNfs [45] introduces a large-scale external molecular library as context to augment the limited known information. GS-Meta [73] further incorporates auxiliary task to depict the many-to-many relationships.

**Parameter-efficient tuning.** As pre-training techniques have advanced, tuning of pre-trained models has become increasingly crucial. Traditional full fine-tuning approaches updates all parameters, often leading to high computational costs and the risk of over-fitting, especially when available data for downstream tasks are limited [33, 15]. This challenge has led to the emergence of parameter-efficient tuning [26, 29, 34]. The philosophy of parameter-efficient tuning is to optimize a small subset of parameters, reducing the computational costs while retaining or even improving performance on downstream tasks [19, 69]. Among the various strategies, the adapters [18, 42, 59] have gained prominence. Adapters are small modules inserted between the pre-trained layers. During the tuning process, only the parameters of these adapters are updated while the rest remains frozen, which not only improves tuning efficiency but also offers an elegant solution to the generalization [70, 30, 8]. By keeping the majority of the pre-trained parameters intact, adapters preserve the rich pre-trained knowledge. This attribute is particularly valuable in many real-world applications including FSMPP.

## 3   Preliminaries

### 3.1   Problem formulation

Let $\{\mathcal{T}\}$ be a collection of tasks, where each task $\mathcal{T}$ involves the prediction of a property $p$. The training set comprising multiple tasks $\{\mathcal{T}_{\text{train}}\}$, is represented as $\mathcal{D}_{\text{train}} = \{(m_i, y_{i,t}) | t \in \{\mathcal{T}_{\text{train}}\}\}$, with $m_i$ indicating a molecule and $y_{i,t}$ its associated label for task $t$. Correspondingly, the test set $\mathcal{D}_{\text{test}}$, formed by tasks $\{\mathcal{T}_{\text{test}}\}$, ensures a separation of properties between training and testing phases, as the property sets $\{p_{\text{train}}\}$ and $\{p_{\text{test}}\}$ are disjoint ($\{p_{\text{train}}\} \cap \{p_{\text{test}}\} = \emptyset$).

The goal of FSMPP is to train a model using $\mathcal{D}_{\text{train}}$ that can accurately infer new properties from a limited number of labeled molecules in $\mathcal{D}_{\text{test}}$. Episodic training has emerged as a promising strategy in meta-learning [10, 16] to deal with few-shot problem. Instead of retaining all $\{\mathcal{T}_{\text{train}}\}$ tasks in memory, episodes $\{E_t\}_{t=1}^{B}$ are iteratively sampled throughout the training process. For each episode $E_t$, a particular task $\mathcal{T}_t$ is selected from the training set, along with corresponding support set $\mathcal{S}_t$ and query set $\mathcal{Q}_t$. Typically, the prediction task involves classifying molecules into two classes: *positive* ($y = 1$) or *negative* ($y = 0$). Then a 2-way $K$-shot episode $E_t = (\mathcal{S}_t, \mathcal{Q}_t)$ is constructed. The support set $\mathcal{S}_t = \{(m_i^s, y_{i,t}^s)\}_{i=1}^{2K}$ includes $2K$ examples, each class contributing $K$ molecules. The query set containing $M$ molecules is denoted as $\mathcal{Q}_t = \{(m_i^q, y_{i,t}^q)\}_{i=1}^{M}$.

### 3.2   Encoder-classifier framework for FSMPP

Encoder-classifier framework is widely adopted in FSMPP methods. As illustrated in Figure 2(a), given a molecule $m$ whose property need to be predicted, a molecular encoder $f(\cdot)$ first learns the molecule's representation based on its structure, i.e., $\boldsymbol{h}_m = f(m) \in \mathbb{R}^d$. The molecule $m$ is generally represented as a graph $m = (\mathcal{V}, \mathbf{A}, \mathbf{X}, \mathbf{E})$, where $\mathcal{V}$ denotes the nodes (atoms), $\mathbf{A}$ represents the adjacent matrix defined by edges (chemical bonds), and $\mathbf{X}, \mathbf{E}$ denote the original feature of atoms

and bonds, then graph neural networks (GNNs) are employed as the molecular encoders [44, 51, 21]. Subsequently, the learned molecular representation is fed into a classifier $g(\cdot)$ to obtain the prediction $\hat{y} = g(\boldsymbol{h}_m)$. The model is trained by minimizing the discrepancy between $\hat{y}$ and the ground truth $y$.

Further, two key discoveries have been pivotal for FSMPP. The first is the proven effectiveness of pre-trained molecular encoders, while the second is the significant advantage gained from molecular context. Together, these discoveries have further reshaped the widely adopted FSMPP framework, which combines a *pre-trained encoder* followed by a *context-aware classifier*, as shown in Figure 2(b).

### 3.3 Pre-trained molecular encoders (PMEs)

Due to the scarcity of labeled data in molecular tasks, molecular pre-training has emerged as a crucial area, which involves training encoders on extensive molecular datasets to extract informative representations. Pre-GNN [20] is a classic pre-trained molecular encoder that has been widely used in addressing FSMPP tasks [14, 58, 73]. The backbone of Pre-GNN is a modified version of Graph Isomorphism Network (GIN) [65] tailored to molecules, which we call GIN-Mol, consisting of multiple atom/bond embedding layers and message passing layers.

**Atom/Bond embedding layers.** The raw atom features and bond features are both categorical vectors, denoted as $(i_{v,1}, i_{v,2}, \ldots, i_{v,|E_n|})$ and $(j_{e,1}, j_{e,2}, \ldots, j_{e,|E_e|})$ for atom $v$ and bond $e$, respectively. These categorical features are embedded as:

$$\boldsymbol{h}_v^{(0)} = \sum_{a=1}^{|E_n|} \texttt{EmbAtom}_a(i_{v,a}), \quad \boldsymbol{h}_e^{(l)} = \sum_{b=1}^{|E_e|} \texttt{EmbBond}_b^{(l)}(j_{e,b}), \tag{1}$$

where $\texttt{EmbAtom}_a(\cdot)_{a \in \{1,\ldots,|E_n|\}}$ and $\texttt{EmbBond}_b(\cdot)_{b \in \{1,\ldots,|E_e|\}}$ represent embedding operations that map integer indices to $d$-dimensional real vectors, i.e., $\boldsymbol{h}_v^{(0)}, \boldsymbol{h}_e^{(l)} \in \mathbb{R}^d$, $l \in \{0, 1, \ldots, L-1\}$ represents the index of encoder layers, and $L$ is the number of encoder layers. The atom embedding layer is present only in the first encoder layer, while an bond embedding layer exists in each layer.

**Message passing layers.** At the $l$-th encoder layer, atom representations are updated by aggregating the features of neighboring atoms and chemical bonds:

$$\boldsymbol{h}_v^{(l)} = \texttt{ReLU}\left(\texttt{MLP}^{(l)}\left(\sum_u \boldsymbol{h}_u^{(l-1)} + \sum_{e=(v,u)} \boldsymbol{h}_e^{(l-1)}\right)\right), \tag{2}$$

where $u \in \mathcal{N}(v) \cup \{v\}$ is the set of atoms connected to $v$, and $\boldsymbol{h}_v^{(l)} \in \mathbb{R}^d$ is the learned representation of atom $v$ at the $l$-th layer. $\texttt{MLP}(\cdot)$ is implemented by 2-layer neural networks, in which the hidden dimension is $d_1$. After $\texttt{MLP}$, batch normalization is applied right before the $\texttt{ReLU}$. The molecule-level representation $\boldsymbol{h}_m \in \mathbb{R}^d$ is obtained by averaging the atom representations at the final layer.

## 4 The proposed Pin-Tuning method

This section delves into our motivation and proposed method. Our framework for FSMPP is depicted in Figure 2(c). The details of our principal design, $\texttt{Pin-Tuning}$ for PMEs, is present in Figure 2(d).

As shown in Figure 1, pretraining then finetuning molecular encoders is a common approach. However, fully fine-tuning yields results inferior to simply freezing them. Thus, the following question arises:

> *How to effectively adapt pre-trained molecular encoders to downstream tasks, especially in few-shot scenarios?*

We analyze the reasons of observed ineffective fine-tuning issue, and attribute it to two primary factors: (i) imbalance between the abundance of tunable parameters and the scarcity of labeled molecules, and (ii) limited contextual perceptiveness in the encoder.

### 4.1 Parameter-efficient tuning for PMEs

To address the first cause of observed ineffective tuning, we reform the tuning method for PMEs. Instead of conducting full fine-tuning for all parameters, we propose tuning strategies specifically tailored to the message passing layers and embedding layers in PMEs, respectively.

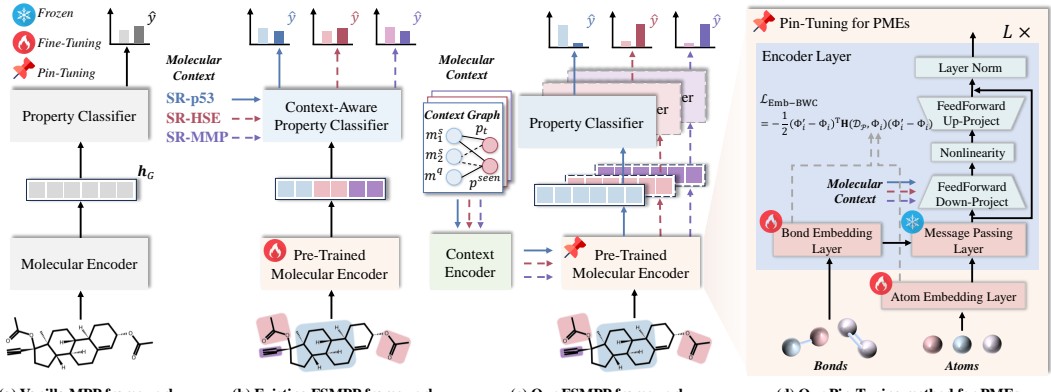

(a) Vanilla MPP framework.  (b) Existing FSMPP framework.  (c) Our FSMPP framework.  (d) Our Pin-Tuning method for PMEs.

Figure 2: (a) The vanilla encoder-classifier framework for MPP. (b) The framework widely adopted by existing FSMPP methods, which contains a pre-trained molecular encoder and a context-aware property classifier. (c) Our proposed framework for FSMPP, in which we introduce a `Pin-Tuning` method to update the pre-trained molecular encoder followed by the property classifier. (d) The details of our proposed Pin-Tuning method for pre-trained molecular encoders. In (b) and (c), we use the property names like SR-HSE to denote the molecular context in episodes.

### 4.1.1 MP-Adapter: message passing layer-oriented adapter

For message passing layers in PMEs, the number of parameters is disproportionately large compared to the training samples. To mitigate this imbalance, we design a lightweight adapter targeted at the message passing layers, called `MP-Adapter`. The pre-trained parameters in each message passing layer include parameters in the MLP and the following batch normalization. We freeze all pre-trained parameters in message passing layers and add a lightweight trainable adapter after MLP in each message passing layer. Formally, the adapter module for $l$-th layer can be represented as:

$$\boldsymbol{z}_v^{(l)} = \texttt{FeedForward}_{\text{down}}(\boldsymbol{h}_v^{(l)}) \in \mathbb{R}^{d_2}, \tag{3}$$

$$\Delta\boldsymbol{h}_v^{(l)} = \texttt{FeedForward}_{\text{up}}(\phi(\boldsymbol{z}_v^{(l)})) \in \mathbb{R}^d, \tag{4}$$

$$\tilde{\boldsymbol{h}}_v^{(l)} = \texttt{LayerNorm}(\boldsymbol{h}_v^{(l)} + \Delta\boldsymbol{h}_v^{(l)}) \in \mathbb{R}^d, \tag{5}$$

where $\texttt{FeedForward}(\cdot)$ denotes feed forward layer and $\texttt{LayerNorm}(\cdot)$ denotes layer normalization. To limit the number of parameters, we introduce a bottleneck architecture. The adapters downscale the original features from $d$ dimensions to a smaller dimension $d_2$, apply nonlinearity $\phi$, then upscale back to $d$ dimensions. By setting $d_2$ smaller than $d$, we can limit the number of parameters added. The adapter module has a skip-connection internally. With the skip-connection, we adopt the near-zero initialization for parameters in the adapter modules, so that the modules are initialized to approximate identity functions. Therefore, the encoder with initialized adapters is equivalent to the pre-trained encoder. Furthermore, we add a layer normalization after skip-connection for training stability.

### 4.1.2 Emb-BWC: embedding layer-oriented Bayesian weight consolidation

Unlike message passing layers, embedding layers contain fewer parameters. Therefore, we directly fine-tune the parameters of the embedding layers, but impose a constraint to limit the magnitude of parameter updates, preventing aggressive optimization and catastrophic forgetting.

The parameters in an embedding layer consist of an embedding matrix used for lookups based on the indices of the original features. We stack the embedding matrices of all embedding layers to form $\Phi \in \mathbb{R}^{E \times d}$, where $E$ represents the total number of lookup entries. Further, $\Phi_i \in \mathbb{R}^d$ denotes the $i$-th row's embedding vector, and $\Phi_{i,j} \in \mathbb{R}$ represents the $j$-th dimensional value of $\Phi_i$.

To avoid aggressive optimization of $\Phi$, we derive a Bayesian weight consolidation framework tailored for embedding layers, called `Emb-BWC`, by applying Bayesian learning theory [3] to fine-tuning.

**Proposition 1:** *(`Emb-BWC` ensures an appropriate stability-plasticity trade-off for pre-trained embedding layers.) Let $\Phi \in \mathbb{R}^{E \times d}$ be the pre-trained embeddings before fine-tuning, and $\Phi' \in \mathbb{R}^{E \times d}$ be the fine-tuned embeddings. Then, the embeddings can both retain the atom and bond properties*

*obtained from pre-training and be appropriately updated to adapt to downstream FSMPP tasks, by introducing the following Emb-BWC loss into objective during the fine-tuning process:*

$$\mathcal{L}_{\text{Emb-BWC}} = -\frac{1}{2} \sum_{i=1}^{E} (\Phi_i' - \Phi_i)^\top \mathbf{H}(\mathcal{D}_\mathcal{P}, \Phi_i)(\Phi_i' - \Phi_i), \tag{6}$$

*where $\mathbf{H}(\mathcal{D}_\mathcal{P}, \Phi_i) \in \mathbb{R}^{d \times d}$ is the Hessian of the log likelihood $\mathcal{L}_\mathcal{P}$ of pre-training dataset $\mathcal{D}_\mathcal{P}$ at $\Phi_i$.*

Details on the theoretical derivation of Eq. (6) are given in Appendix A. Since $\mathbf{H}(\mathcal{D}_\mathcal{P}, \Phi_i)$ is intractable to compute due to the great dimensionality of $\Phi$, we adopt the diagonal approximation of Hessian. By approximating $\mathbf{H}$ as a diagonal matrix, the $j$-th value on the diagonal of $\mathbf{H}$ can be considered as the importance of the parameter $\Phi_{i,j}$. The following three choices are considered.

***Identity matrix.*** When using the identity matrix to approximate the negation of $\mathbf{H}$, Eq. (6) is simplified to $\mathcal{L}_{\text{Emb-BWC}}^{\text{IM}} = \frac{1}{2} \sum_{i=1}^{E} \sum_{j=1}^{d} (\Phi_{i,j}' - \Phi_{i,j})^2$, assigning equal importance to each parameter. This loss function is also known as L2 penalty with pre-trained model as the starting point (L2-SP) [31].

***Diagonal of Fisher information matrix.*** The Fisher information matrix (FIM) $\mathbf{F}$ is the negation of the expectation of the Hessian over the data distribution, i.e., $\mathbf{F} = -\mathbb{E}_{\mathcal{D}_\mathcal{P}}[\mathbf{H}]$, and the FIM can be further simplified with a diagonal approximation. Then, the Eq. (6) is simplified to $\mathcal{L}_{\text{Emb-BWC}}^{\text{FIM}} = \frac{1}{2} \sum_{i=1}^{E} \hat{\mathbf{F}}_i (\Phi_i' - \Phi_i)^2$, where $\hat{\mathbf{F}}_i \in \mathbb{R}^d$ is the diagonal of $\mathbf{F}(\mathcal{D}_\mathcal{P}, \Phi_i) \in \mathbb{R}^{d \times d}$ and the $j$-th value in $\hat{\mathbf{F}}_i$ is computed as $\mathbb{E}_{\mathcal{D}_\mathcal{P}}(\partial \mathcal{L}_\mathcal{P} / \partial \Phi_{i,j})^2$. This is equivalent to elastic weight consolidation (EWC) [24].

***Diagonal of embedding-wise Fisher information matrix.*** In different property prediction tasks, the impact of the same atoms and inter-atomic interactions may be significant or negligible. Therefore, we propose this choice to assign importance to parameters based on different embeddings, rather than treating each parameter individually. By defining $\tilde{\Phi}_i = \sum_j \Phi_{i,j}$, the total update of the embedding $\Phi_i$ can be represented as $\Delta \Phi_i = \tilde{\Phi}_i' - \tilde{\Phi}_i = \sum_j (\Phi_{i,j}' - \Phi_{i,j})$. Then, the Eq. (6) is reformulated to $\mathcal{L}_{\text{Emb-EWC}}^{\text{EFIM}} = \frac{1}{2} \sum_{i=1}^{E} \tilde{\mathbf{F}}_i (\tilde{\Phi}_i' - \tilde{\Phi}_i)^2$, where $\tilde{\mathbf{F}}_i = \sum_j \mathbb{E}_{\mathcal{D}_\mathcal{P}}(\partial \mathcal{L}_\mathcal{P} / \partial \Phi_{i,j})^2$.

Detailed derivation is given in Appendix A. Intuitively, these three approximations employ different methods to assign importance to parameters. $\mathcal{L}_{\text{Emb-BWC}}^{\text{IM}}$ assigns the same importance to each parameter, $\mathcal{L}_{\text{Emb-BWC}}^{\text{FIM}}$ assigns individual importance to each parameter, and $\mathcal{L}_{\text{Emb-BWC}}^{\text{EFIM}}$ assigns the same importance to parameters within the same embedding vector.

## 4.2 Enabling contextual perceptiveness in MP-Adapter

For different property prediction tasks, the decisive substructures vary. As shown in Figure 2, the ester group in the given molecule determines the property SR-HSE, while the carbon-carbon triple bond determines the property SR-MMP. If fine-tuning can be guided by molecular context, encoding context-specific molecular representations allows for dynamic representations of molecules tailored to specific tasks and enables the modeling of the context-specific significance of substructures.

**Extracting molecular context information.** In each episode, we consider the labels of the support molecules on the target property and seen properties, as well as the labels of the query molecules on seen properties, as the context of this episode. We adopt the form of a graph to describe the context. Figure 3 demonstrates the transformation from original context data to a context graph. In the left table, the labels of molecules $m_1^q, m_2^q$ for property $p_t$ are the prediction targets, and the other shaded values are the available context. The right side shows the

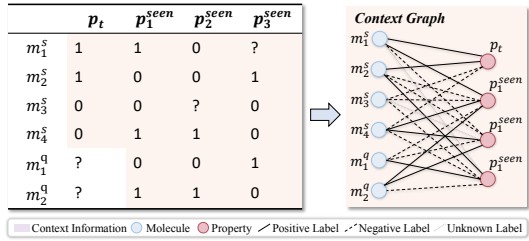

Figure 3: Convert the context information of a 2-shot episode into a context graph.

context graph constructed based on the available context. Specifically, we construct context graph $\mathcal{G}_t = (\mathcal{V}_t, \mathbf{A}_t, \mathbf{X}_t)$ for episode $E_t$. It contains $M$ molecule nodes $\{m\}$ and $P$ property nodes $\{p\}$. Three types of edges indicate different relationships between molecules and properties.

Then we employ a GNN-based context encoder: $\mathbf{C} = \texttt{ContextEncoder}(\mathcal{V}_t, \mathbf{A}_t, \mathbf{X}_t)$, where $\mathbf{C} \in \mathbb{R}^{(M+P) \times d_2}$ denotes the learned context representation matrix for $E_t$. $\mathcal{V}_t$ and $\mathbf{A}_t$ denote the node set

and the adjacent matrix of the context graph, respectively, and $\mathbf{X}_t$ denotes the initial features of nodes. The features of molecule nodes are initialized with a pre-trained molecular encoder. The property nodes are randomly initialized. When we make the prediction of molecule $m$'s target property $p$, we take the learned representations of the this molecule $c_m$ and of the target property $c_p$ as the context vectors. Details about the context encoder are provided in Appendix F.2.

**In-context tuning with molecular context information.** After obtaining the context vectors, we consider enabling the molecular encoder to use the context as a condition, achieving conditional molecular encoding. To achieve this, we further refine our adapter module. While neural conditional encoding has been explored in some domains, such as cross-attention [43] and ControlNet [68] for conditional image generation, these methods often come with a significant increase in the number of parameters. This contradicts our motivation of parameter-efficient tuning for few-shot tasks. In this work, we adopt a simple yet effective method. We directly concatenate the context with the output of the message passing layer, and feed them into the downscaling feed-forward layer in the `MP-Adapter`. Formally, the downscaling process defined in Eq. (3) is reformulated as:

$$z^{(l)} = \texttt{FeedForward}_{\texttt{down}}(h_v^{(l)} \| c_m \| c_p), \tag{7}$$

where $\|$ denotes concatenation. Such learned molecular representations are more easily predicted on specific properties, verified in Section 5.5 and Appendix G.

## 4.3 Optimization

Following MAML [10], a gradient descent strategy is adopted. Firstly, $B$ episodes $\{E_t\}_{t=1}^B$ are randomly sampled. For each episode, in the inner-loop optimization, the loss on the support set is computed as $\mathcal{L}_{t,\mathcal{S}}^{cls}(f_\theta)$ and the parameters $\theta$ are updated by gradient descent:

$$\mathcal{L}_{t,\mathcal{S}}^{cls}(f_\theta) = -\sum\nolimits_{\mathcal{S}_t} (y \log(\hat{y}) + (1-y)\log(1-\hat{y})), \tag{8}$$

$$\theta' \leftarrow \theta - \alpha_{inner} \nabla_\theta \mathcal{L}_{t,\mathcal{S}}^{cls}(f_\theta), \tag{9}$$

where $\alpha_{inner}$ is the learning rate. In the outer loop, the classification loss of query set is denoted as $\mathcal{L}_{t,Q}^{cls}$. Together with our `Emb-BWC` regularizer, the meta-training loss $\mathcal{L}(f_{\theta'})$ is computed and we do an outer-loop optimization with learning rate $\alpha_{outer}$ across the mini-batch:

$$\mathcal{L}(f_{\theta'}) = \frac{1}{B} \sum\nolimits_{t=1}^B \mathcal{L}_{t,\mathcal{Q}}^{cls}(f_{\theta'}) + \lambda \mathcal{L}_{\text{Emb-BWC}}, \tag{10}$$

$$\theta \leftarrow \theta - \alpha_{outer} \nabla_\theta \mathcal{L}(f_{\theta'}), \tag{11}$$

where $\lambda$ is the weight of `Emb-BWC` regularizer. The pseudo-code is provided in Appendix B. We also provide more discussion of tunable parameter size and total model size in  Appendix C.

# 5 Experiments

## 5.1 Evaluation setups

**Datasets.** We use five common few-shot molecular property prediction datasets from the MoleculeNet [61]: Tox21, SIDER, MUV, ToxCast, and PCBA. Standard data splits for FSMPP are adopted. Dataset statistics and more details of datasets can be found in Appendix D.

**Baselines.** For a comprehensive comparison, we adopt two types of baselines: (1) methods with molecular encoders trained from scratch, including Siamese Network [25], ProtoNet [46], MAML [10], TPN [35], EGNN [22], and IterRefLSTM [1]; and (2) methods which leverage pre-trained molecular encoders, including Pre-GNN [20], Meta-MGNN [14], PAR [58], and GS-Meta [73]. More details about these baselines are in Appendix E.

**Metrics.** Following prior works [1, 58], ROC-AUC scores are calculated on the query set for each meta-testing task, to evaluate the performance of FSMPP. We run experiments 10 times with different random seeds and report the mean and standard deviations.

Table 1: ROC-AUC scores (%) on benchmark datasets, compared with methods trained from scratch (first group) and methods that leverage pre-trained molecular encoder (second group). The best is marked with **boldface** and the second best is with underline. $\Delta$*Improve.* indicates the relative improvements over the baseline models in percentage.

| Model | Tox21 10-shot | Tox21 5-shot | SIDER 10-shot | SIDER 5-shot | MUV 10-shot | MUV 5-shot | ToxCast 10-shot | ToxCast 5-shot | PCBA 10-shot | PCBA 5-shot |
|---|---|---|---|---|---|---|---|---|---|---|
| Siamese | 80.40(0.35) | - | 71.10(4.32) | - | 59.96(5.13) | - | - | - | - | - |
| ProtoNet | 74.98(0.32) | 72.78(3.93) | 64.54(0.89) | 64.09(2.37) | 65.88(4.11) | 64.86(2.31) | 68.87(0.43) | 66.26(1.49) | 64.93(1.94) | 62.29(2.12) |
| MAML | 80.21(0.24) | 69.17(1.34) | 70.43(0.76) | 60.92(0.65) | 63.90(2.28) | 63.00(0.61) | 68.30(0.59) | 67.56(1.53) | 66.22(1.31) | 65.25(0.75) |
| TPN | 76.05(0.24) | 75.45(0.95) | 67.84(0.95) | 66.52(1.28) | 65.22(5.82) | 65.13(0.23) | 69.47(0.71) | 66.04(1.14) | 67.61(0.33) | 63.66(1.64) |
| EGNN | 81.21(0.16) | 76.80(2.62) | 72.87(0.73) | 60.61(1.06) | 65.20(2.08) | 63.46(2.58) | 74.02(1.11) | 67.13(0.50) | 69.92(1.85) | 67.71(3.67) |
| IterRefLSTM | 81.10(0.17) | - | 69.63(0.31) | - | 49.56(5.12) | - | - | - | - | - |
| Pre-GNN | 82.14(0.08) | 82.04(0.30) | 73.96(0.08) | 76.76(0.53) | 67.14(1.58) | 70.23(1.40) | 75.31(0.95) | 74.43(0.47) | 76.79(0.45) | 75.27(0.49) |
| Meta-MGNN | 82.97(0.10) | 76.12(0.23) | 75.43(0.21) | 66.60(0.38) | 68.99(1.84) | 64.07(0.56) | 76.27(0.56) | 75.26(0.43) | 72.58(0.34) | 72.51(0.52) |
| PAR | 84.93(0.11) | 83.95(0.15) | 78.08(0.16) | 77.70(0.34) | 69.96(1.37) | 68.08(2.42) | 79.41(0.08) | 76.89(0.32) | 73.71(0.61) | 72.79(0.98) |
| GS-Meta | 86.67(0.41) | 86.43(0.02) | 84.36(0.54) | 84.57(0.01) | 66.08(1.25) | 64.50(0.20) | 83.81(0.16) | 82.65(0.35) | 79.40(0.43) | 77.47(0.29) |
| Pin-Tuning | **91.56**(2.57) | **90.95**(2.33) | **93.41**(3.52) | **92.02**(3.01) | **73.33**(2.00) | **70.71**(1.42) | **84.94**(1.09) | **83.71**(0.93) | **81.26**(0.46) | **79.23**(0.52) |
| $\Delta$*Improve.* | 5.64% | 5.23% | 10.73% | 8.81% | 4.82% | 3.86% | 1.35% | 1.28% | 2.34% | 2.27% |

Table 2: Ablation analysis on the `MP-Adapter`, in which we drop different components to form variants. We report ROC-AUC scores (%), and the best performance is highlighted in **bold**.

| Model | Adapter | Context | LayerNorm | Tox21 10-shot | Tox21 5-shot | SIDER 10-shot | SIDER 5-shot | MUV 10-shot | MUV 5-shot | ToxCast 10-shot | ToxCast 5-shot | PCBA 10-shot | PCBA 5-shot |
|---|---|---|---|---|---|---|---|---|---|---|---|---|---|
| Pin-Tuning | ✓ | ✓ | ✓ | **91.56** | **90.95** | **93.41** | **92.02** | **73.33** | **70.71** | **84.94** | **83.71** | **81.26** | **79.23** |
| w/o Adapter | - | - | ✓ | 79.72 | 78.49 | 74.04 | 72.94 | 66.06 | 62.88 | 80.06 | 78.70 | 73.85 | 72.02 |
| w/o Context | ✓ | - | ✓ | 81.42 | 79.34 | 74.68 | 72.86 | 68.70 | 66.12 | 81.49 | 79.85 | 74.69 | 72.46 |
| w/o LayerNorm | ✓ | ✓ | - | 86.71 | 84.93 | 91.50 | 90.76 | 70.26 | 67.42 | 83.52 | 82.55 | 80.07 | 78.23 |

## 5.2 Performance comparison

We compare `Pin-Tuning` with the baselines and the results are summarized in Table 1, Table 7, and Table 8. Our method significantly outperforms all baseline models under both the 10-shot and 5-shot settings, demonstrating the effectiveness and superiority of our approach.

Across all datasets, our method provides greater improvement in the 10-shot scenario than in the 5-shot scenario. This is attributed to the molecular context constructed based on support molecules. When there are more molecules in the support set, the uncertainty in the context is reduced, providing more effective adaptation guidance for our parameter-efficient tuning.

Among benchmark datasets, our method shows significant improvement on the SIDER dataset, increasing by 10.73% in the 10-shot scenario and by 8.81% in the 5-shot scenario. We consider this is related to the relatively balanced ratio of positive to negative samples, as well as the absence of missing labels in the SIDER dataset (Table 5). A balanced and low-uncertainty distribution can better benefit addressing the FSMPP task from our method.

We also observe that the standard deviations of our method's results under 10 seeds are slightly higher than that of baseline models. However, our worst-case results are still better than the best baseline model. For example, in 10-shot experiments on the Tox21 dataset, the performance of our method is $91.56 \pm 2.57$. However, our 10 runs yield specific results with the worst-case ROC-AUC reaching 88.02, which is also better than the best baseline model GS-Meta's result of $86.67 \pm 0.41$. Therefore, a high standard deviation does not mean our method is inferior to baseline models.

## 5.3 Ablation study

For `MP-Adapter`, the main components consist of: (i) bottleneck adapter module (Adapter), (ii) introducing molecular context to adatpers (Context), and (iii) layer normalization (LayerNorm). The results of ablation experiments are summarized in Table 2. The bottleneck adapter and the modeling of molecular context are the most critical, having the most significant impact on performance. Removing them leads to a noticeable decline, which underscores the importance of parameter-efficient tuning and context perceptiveness in FSMPP tasks. Layer normalization is used to normalize the resulting representations, which is also important for improving the optimization effect and stability.

For `Emb-BWC`, we verify the effectiveness of fine-tuning the embedding layers and regularizing them with different approximations of $\mathcal{L}_{\text{Emb-BWC}}$ (Table 3). Since the embedding layers have relatively few parameters, direct fine-tuning can also enhance performance. Applying our proposed regularizers to fine-tuning can further improve the effects. Among the three regularizers, the

Table 3: Ablation analysis on the `Emb-BWC`.

| Fine-tune | Regularizer | Tox21 | SIDER | MUV | PCBA |
|---|---|---|---|---|---|
| - | - | 89.70 | 90.12 | 70.76 | 80.24 |
| ✓ | - | 90.17 | 92.06 | 72.37 | 80.74 |
| ✓ | $\mathcal{L}_{\text{Emb-BWC}}^{\text{IM}}$ | **91.56** | **93.41** | **73.22** | **81.26** |
| ✓ | $\mathcal{L}_{\text{Emb-BWC}}^{\text{FIM}}$ | 90.93 | 90.09 | 72.17 | 80.78 |
| ✓ | $\mathcal{L}_{\text{Emb-BWC}}^{\text{EFIM}}$ | 91.32 | 90.31 | 72.78 | 81.22 |

$\mathcal{L}_{\text{Emb-BWC}}^{\text{IM}}$ is the most effective. This indicates that keeping pre-trained parameters to some extent can better utilize pre-trained knowledge, but the parameters worth keeping in fine-tuning and the important parameters in pre-training revealed by Fisher information matrix are not completely consistent.

## 5.4 Sensitivity analysis

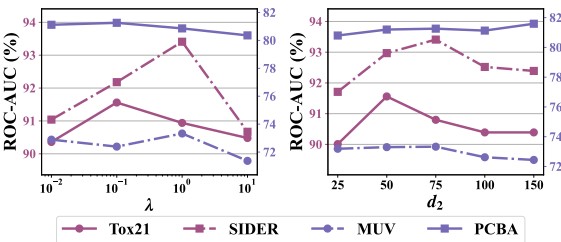 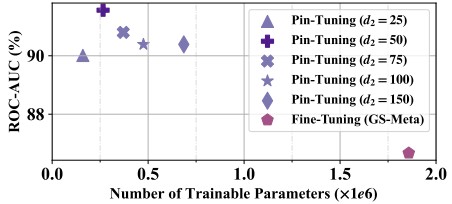

Figure 4: Effect of different hyper-parameters. The y-axis represents ROC-AUC scores (%) and the x-axis is the different hyper-parameters.

Figure 5: ROC-AUC (%) and number of trainable parameters of Pin-Tuning with varied value of $d_2$ and full Fine-Tuning method (e.g., GS-Meta) on the Tox21 dataset.

**Effect of weight of `Emb-BWC` regularizer $\lambda$.** `Emb-BWC` is applied on the embedding layers to limit the magnitude of parameter updates during fine-tuning. We vary the weight of this regularization $\lambda$ from $\{0.01, 0.1, 1, 10\}$. The first subfigure in Figure 4 shows that the performance is best when $\lambda = 0.1$ or 1. When $\lambda$ is too small, the parameters undergo too large updates on few-shot downstream datasets, leading to over-fitting and ineffectively utilizing the pre-trained knowledge. Too large $\lambda$ causes the parameters of the embedding layers to be nearly frozen, which prevents effective adaptation.

**Effect of hidden dimension of `MP-Adapter` $d_2$.** The results corresponding to different values of $d_2$ from $\{25, 50, 75, 100, 150\}$ are presented in the second subfigure of Figure 4. On the Tox21 dataset, we further analyze the impact of this hyper-parameter on the number of trainable parameters. As shown in Figure 5, the number of parameters that our method needs to train is significantly less than that required by the full fine-tuning method, such as GS-Meta, while our method also performs better in terms of ROC-AUC performance due to solving over-fitting and context perceptiveness issues. When $d = 50$, Pin-Tuning performs best on Tox21, and the number of parameters that need to train is only 14.2% of that required by traditional fine-tuning methods.

## 5.5 Case study

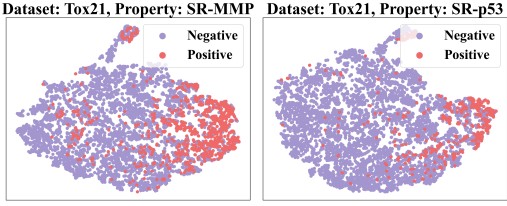 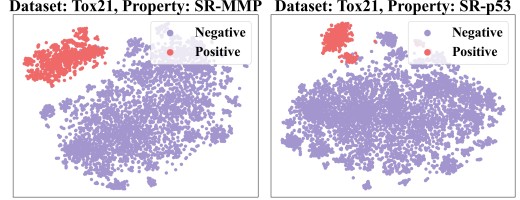

Figure 6: Molecular representations encoded by GS-Meta [73].

Figure 7: Molecular representations encoded by `Pin-Tuning`.

We visualized the molecular representations learned by the GS-Meta and our `Pin-Tuning`'s encoders in the 10-shot setting, respectively. As shown in Figure 6 and 7, `Pin-Tuning` can effectively adapt to different downstream tasks based on context information, generating property-specific molecular representations. Across different tasks, our method is more effective in encoding representations that

facilitate the prediction of the current property, reducing the difficulty of property prediction from the encoding representation aspect. More case studies are provided in Appendix G.

## 6 Conclusion

In this work, we propose a tuning method, `Pin-Tuning`, to address the ineffective fine-tuning of pre-trained molecular encoders in FSMPP tasks. Through the innovative parameter-efficient tuning and in-context tuning for pre-trained molecular encoders, our approach not only mitigates the issues of parameter-data imbalance but also enhances contextual perceptiveness. The promising results on public datasets underscore the potential of `Pin-Tuning` to advance this field, offering valuable insights for future research in drug discovery and material science.

## Acknowledgments

This work is jointly supported by National Science and Technology Major Project (2023ZD0120901), National Natural Science Foundation of China (62372454, 62236010) and the Excellent Youth Program of State Key Laboratory of Multimodal Artificial Intelligence Systems.

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

# Appendix

The organization of the appendix is as follows:

- Appendix A: Derivation of `Emb-BWC` regularization;
- Appendix B: Pseudo-code of training process;
- Appendix C: Discussion of tunable parameter size and total model size;
- Appendix D: Details of datasets;
- Appendix E: Details of baselines;
- Appendix F: Implementation details;
- Appendix G: More experimental results and discussions;
- Appendix H: Limitations and future directions.

## A  Derivation of `Emb-BWC` regularization

### A.1  Derivation of $\mathcal{L}_{\textbf{Emb-BWC}}$

Let $\Phi \in \mathbb{R}^{E \times d}$ be the pre-trained embeddings before fine-tuning, and $\Phi' \in \mathbb{R}^{E \times d}$ be the fine-tuned embeddings. Further, $\Phi_i \in \mathbb{R}^d$ denotes the $i$-th row's embedding vector in $\Phi$, and $\Phi_{i,j} \in \mathbb{R}$ represents the $j$-th dimensional value of $\Phi_i$.

The optimization of embedding layers can be interpreted as performing a maximum a posterior (MAP) estimation of the parameters $\Phi'$ given the pre-training data and training data of downstream FSMPP task, which is formulated in a Bayesian framework.

In the FSMPP setting, the molecular encoder has been pre-trained on the pre-training task $\mathcal{P}$ using data $\mathcal{D}_\mathcal{P}$, and is then fine-tuned on a downstream FSMPP task $\mathcal{F}$ using data $\mathcal{D}_\mathcal{F}$. The overall objective is to find the optimal parameters on task $\mathcal{F}$ while preserving the prior knowledge obtained in pre-training on task $\mathcal{F}$. Based on a prior $p(\Phi')$ of the embedding parameters, the posterior after observing the FSMPP task $\mathcal{F}$ can be computed with Bayes' rule:

$$
\begin{aligned}
p(\Phi'|\mathcal{D}_\mathcal{P}, \mathcal{D}_\mathcal{F}) &= \frac{p(\mathcal{D}_\mathcal{F}|\Phi', \mathcal{D}_\mathcal{P})p(\Phi'|\mathcal{D}_\mathcal{P})}{p(\mathcal{D}_\mathcal{F}|\mathcal{D}_\mathcal{P})} \\
&= \frac{p(\mathcal{D}_\mathcal{F}|\Phi')p(\Phi'|\mathcal{D}_\mathcal{P})}{p(\mathcal{D}_\mathcal{F})},
\end{aligned}
\tag{12}
$$

where $\mathcal{D}_\mathcal{F}$ is assumed to be independent of $\mathcal{D}_\mathcal{P}$. Taking a logarithm of the posterior, the MAP objective is therefore:

$$
\begin{aligned}
\Phi'^* &= \arg\max_{\Phi'} \log p(\Phi'|\mathcal{D}_\mathcal{P}, \mathcal{D}_\mathcal{F}) \\
&= \arg\max_{\Phi'} \log p(\mathcal{D}_\mathcal{F}|\Phi') + \log p(\Phi'|\mathcal{D}_\mathcal{P}).
\end{aligned}
\tag{13}
$$

The first term $\log p(\mathcal{D}_\mathcal{F}|\Phi')$ is the log likelihood of the data $\mathcal{D}_\mathcal{F}$ given the parameters $\Phi'$, which can be expressed as the training loss function on task $\mathcal{F} = -\log p(\mathcal{D}_\mathcal{F}|\Phi')$, denoted as $\mathcal{L}_\mathcal{F}(\Phi')$. The second term $p(\Phi'|\mathcal{D}_\mathcal{P})$ is the posterior of the parameters given the pre-training dataset $\mathcal{D}_\mathcal{P}$. Since $\Phi' = [\Phi_1'^\top, \Phi_2'^\top, \ldots, \Phi_E'^\top]^\top$, and $\Phi_i'$ is conditionally independent of $\Phi_j'$ for $i, j = \{1, \ldots, E\}$ and $i \neq j$ given condition $\mathcal{D}_\mathcal{P}$, we have $p(\Phi'|\mathcal{D}_\mathcal{P}) = \prod_{i=1}^E p(\Phi_i'|\mathcal{D}_\mathcal{P})$. Thus, $\log p(\Phi'|\mathcal{D}_\mathcal{P}) = \sum_{i=1}^E \log p(\Phi_i'|\mathcal{D}_\mathcal{P})$

For adapting pre-trained molecular embedding layers to downstream FMSPP tasks, this posterior must encompass the prior knowledge of the pre-trained embedding layers to reflect which parameters are important for pre-training task $\mathcal{P}$. Despite the true posterior being intractable, $\log p(\Phi_i'|\mathcal{D}_\mathcal{P})$ can be defined as a function $f(\Phi_i')$ and approximated around the optimum point $f(\Phi_i)$, where $f(\Phi_i)$ is the pre-trained values and $\nabla f(\Phi_i) = 0$. Performing a second-order Taylor expansion on $f(\Phi_i')$ around $\Phi_i$ gives:

$$
\begin{aligned}
\log p(\Phi_i' \mid \mathcal{D}_\mathcal{P}) &\approx f(\Phi_i) + \frac{1}{2}(\Phi_i' - \Phi_i)^T \nabla^2 f(\Phi_i)(\Phi_i' - \Phi_i) \\
&= f(\Phi_i) + \frac{1}{2}(\Phi_i' - \Phi_i)^T \mathbf{H}(\mathcal{D}_\mathcal{P}, \Phi_i)(\Phi_i' - \Phi_i),
\end{aligned}
\tag{14}
$$

where $\mathbf{H}(\mathcal{D}_\mathcal{P}, \Phi_i) \in \mathbb{R}^{d \times d}$ is the Hessian matrix of $f(\Phi_i')$ at $\Phi_i$. The second term suggests that the posterior of the parameters on the pre-training data can be approximated by a Gaussian distribution with mean $\Phi_i'$ and covariance $\mathbf{H}(\mathcal{D}_\mathcal{P}, \Phi_i)^{-1}$. Following Eq. (13), the training objective becomes:

$$
\begin{aligned}
\Phi'^* &= \arg\max_{\Phi'} \log p(\mathcal{D}_\mathcal{F}|\Phi') + \log p(\Phi'|\mathcal{D}_\mathcal{P}) \\
&= \arg\max_{\Phi'} \log p(\mathcal{D}_\mathcal{F}|\Phi') + \sum_{i=1}^{E} \log p(\Phi_i'|\mathcal{D}_\mathcal{P}) \\
&= \arg\min_{\Phi'} \mathcal{L}_\mathcal{F}(\Phi') - \sum_{i=1}^{E} f(\Phi_i) - \frac{1}{2}\sum_{i=1}^{E}(\Phi_i' - \Phi_i)^T \mathbf{H}(\mathcal{D}_\mathcal{P}, \Phi_i)(\Phi_i' - \Phi_i) \\
&= \arg\min_{\Phi'} \mathcal{L}_\mathcal{F}(\Phi') - \frac{1}{2}\sum_{i=1}^{E}(\Phi_i' - \Phi_i)^T \mathbf{H}(\mathcal{D}_\mathcal{P}, \Phi_i)(\Phi_i' - \Phi_i).
\end{aligned}
\tag{15}
$$

We define the second term as our `Emb-BWC` regularization objective:

$$
\mathcal{L}_{\text{Emb-BWC}} = -\frac{1}{2}\sum_{i=1}^{E}(\Phi_i' - \Phi_i)^\top \mathbf{H}(\mathcal{D}_\mathcal{P}, \Phi_i)(\Phi_i' - \Phi_i).
\tag{16}
$$

## A.2 Derivation of $\mathcal{L}_{\text{Emb-BWC}}^{\text{FIM}}$

Since the Fisher information matrix (FIM) $\mathbf{F}$ is the negation of the expectation of the Hessian over the data distribution, i.e., $\mathbf{F} = -\mathbb{E}_{\mathcal{D}_\mathcal{P}}[\mathbf{H}]$, the objective can be reformulated as:

$$
\mathcal{L}_{\text{Emb-BWC}}^{\text{FIM}} = \frac{1}{2}\sum_{i=1}^{E}(\Phi_i' - \Phi_i)^\top \mathbf{F}(\mathcal{D}_\mathcal{P}, \Phi_i)(\Phi_i' - \Phi_i),
\tag{17}
$$

where $\mathbf{F}(\mathcal{D}_\mathcal{P}, \Phi_i) \in \mathbb{R}^{d \times d}$ is the corresponding Fisher information matrix of $\mathbf{H}(\mathcal{D}_\mathcal{P}, \Phi_i)$. Further, the Fisher information matrix can be further simplified with a diagonal approximation. Then, the objective is simplified to:

$$
\mathcal{L}_{\text{Emb-BWC}}^{\text{FIM}} \approx \frac{1}{2}\sum_{i=1}^{E} \hat{\mathbf{F}}_i(\Phi_i' - \Phi_i)^2,
\tag{18}
$$

where $\hat{\mathbf{F}}_i \in \mathbb{R}^d$ is the diagonal of $\mathbf{F}(\mathcal{D}_\mathcal{P}, \Phi_i)$. According to the definition of the Fisher information matrix, the $j$-th value in $\hat{\mathbf{F}}_i$ is computed as $\mathbb{E}_{\mathcal{D}_\mathcal{P}}(\partial \mathcal{L}_\mathcal{P}/\partial \Phi_{i,j})^2$. In this work, this approximated form is defined as $\mathcal{L}_{\text{Emb-BWC}}^{\text{FIM}}$.

## A.3 Derivation of $\mathcal{L}_{\text{Emb-BWC}}^{\text{EFIM}}$

We assume that the parameters within an embedding should share the same importance. To this end, we define $\tilde{\Phi}_i = \sum_j \Phi_{i,j}$, then the total update of the embedding $\Phi_i$ can be represented as $\Delta \Phi_i = \tilde{\Phi}_i' - \tilde{\Phi}_i = \sum_j(\Phi_{i,j}' - \Phi_{i,j})$. Then, the objective in Eq. (16) is reformulated to:

$$
\mathcal{L}_{\text{Emb-EWC}}^{\text{EFIM}} = \frac{1}{2}\sum_{i=1}^{E} \tilde{\mathbf{H}}_i(\tilde{\Phi}_i' - \tilde{\Phi}_i)^2,
\tag{19}
$$

where $\tilde{\mathbf{H}}_i = \frac{\partial^2 \mathcal{L}_\mathcal{P}}{\partial \tilde{\Phi}_i^2} = \frac{\partial}{\partial \tilde{\Phi}_i}\left(\frac{\partial \mathcal{L}_\mathcal{P}}{\partial \tilde{\Phi}_i}\right)$. Next, we continue to derive $\tilde{\mathbf{H}}_i$. Given that $\tilde{\Phi}_i = \sum_{j=1}^{d}\Phi_{i,j}$, we first use the chain rule to find $\frac{\partial \mathcal{L}_\mathcal{P}}{\partial \tilde{\Phi}_i}$ According to chain rule, the derivative of $\mathcal{L}_\mathcal{P}$ with respect to $\partial \tilde{\Phi}_i$ can be computed as:

$$
\frac{\partial \mathcal{L}_\mathcal{P}}{\partial \tilde{\Phi}_i} = \sum_j \frac{\partial \mathcal{L}_\mathcal{P}}{\partial \Phi_{i,j}}\frac{\partial \Phi_{i,j}}{\partial \tilde{\Phi}_i}.
\tag{20}
$$

Since $\tilde{\Phi}_i = \Phi_{i,1} + \Phi_{i,2} + \ldots + \Phi_{i,d}$, each of $\frac{\partial \Phi_{i,j}}{\partial \tilde{\Phi}_i}$ for $j = 1, 2, \ldots, d$ equals 1. Therefore, the equation simplifies to:

$$
\frac{\partial \mathcal{L}_\mathcal{P}}{\partial \tilde{\Phi}_i} = \sum_{j=1}^{d} \frac{\partial \mathcal{L}_\mathcal{P}}{\partial \Phi_{i,j}}.
\tag{21}
$$

When taking the derivative of this with respect to $\Phi_{i,j}$ again, using the chain rule, we have:

$$\frac{\partial^2 \mathcal{L}_\mathcal{P}}{\partial \tilde{\Phi}_i^2} = \frac{\partial}{\partial \tilde{\Phi}_i}\left(\sum_{j=1}^d \frac{\partial \mathcal{L}_\mathcal{P}}{\partial \Phi_{i,j}}\right) = \sum_{j=1}^d \sum_{k=1}^d \frac{\partial}{\partial \Phi_{i,k}}\left(\frac{\partial \mathcal{L}_\mathcal{P}}{\partial \Phi_{i,j}}\right)\frac{\partial \Phi_{i,k}}{\partial \tilde{\Phi}_i} \tag{22}$$

Given that $\Phi_{i,j}(\Phi_{i,k})$ are all parameters in embedding lookup tables, they are independent of each other. Thus, when $j = k$, $\frac{\partial}{\partial \Phi_{i,k}}\left(\frac{\partial \mathcal{L}_\mathcal{P}}{\partial \Phi_{i,j}}\right)\frac{\partial \Phi_{i,k}}{\partial \tilde{\Phi}_i} = \frac{\partial^2 \mathcal{L}_\mathcal{P}}{\partial \Phi_{i,j}^2}$, otherwise it equals 0. We finally get $\tilde{\mathbf{H}}_i = \sum_j \frac{\partial^2 \mathcal{L}_\mathcal{P}}{\partial \Phi_{i,j}^2} = \sum_j \mathbf{H}(\mathcal{D}_\mathcal{P}, \Phi_i)_{j,j}$.

We still approximate the Hessian with the FIM as in Section A.2, and combining this with the definition of FIM, we arrive at the final objective:

$$\mathcal{L}_{\text{Emb-EWC}}^{\text{EFIM}} \approx \frac{1}{2}\sum_{i=1}^E \tilde{\mathbf{F}}_i(\tilde{\Phi}_i' - \tilde{\Phi}_i)^2, \tag{23}$$

where $\tilde{\mathbf{F}}_i = \sum_j \mathbb{E}_{\mathcal{D}_\mathcal{P}}(\partial \mathcal{L}_\mathcal{P}/\partial \Phi_{i,j})^2$.

# B  Pseudo-code of training process

To help better understand the training process, we provide the brief pseudo-code of it in Algorithm 1.

---
**Algorithm 1:** Training process of `Pin-Tuning`.

**Input** : Training set $\mathcal{D}_{\text{train}}$
**Output** : Tuned few-shot molecular property prediction model with parameter $\theta$

1 **while** *not converge* **do**
2    Sample $B$ episode from training set $\mathcal{D}_{\text{train}}$ to form a mini-batch $\{E_t\}_{t=1}^B$;
3    **for** $t = 1$ **to** $B$ **do**
4       Calculate classification loss on support set $\mathcal{L}_{t,\mathcal{S}}^{cls}(f_\theta)$ by Eq. (8) on $E_t$:
         $\theta' \leftarrow \theta - \alpha_{inner}\nabla_\theta \mathcal{L}_{t,\mathcal{S}}^{cls}(f_\theta)$;
5       Do inner-loop update by Eq. (9) on $E_t$;
6       Calculate classification loss on query set $\mathcal{L}_{t,\mathcal{Q}}^{cls}(f_{\theta'})$ by Eq. (8) on $E_t$;
7    Calculate update constraint $\mathcal{L}_{\text{Emb-BWC}}$ by Eq. (6);
8    Do outer-loop optimization by Eq. (10) and Eq. (11): $\theta \leftarrow \theta - \alpha_{outer}\nabla_\theta \mathcal{L}(f_{\theta'})$;
9 Return optimized model parameter $\theta$.

---

# C  Discussion of tunable parameter size and total model size

## C.1  Tunable parameter size of molecular encoder

We compare the tunable parameter size of full fine-tuning and our `Pin-Tuning`. Section 3.3 describes the parameters of the PME, which include those for the embedding layers and the message passing layers. We assume there are $|E_n|$ original node features and $|E_e|$ edge features. Considering there is one node embedding layer and $L$ edge embedding layers, the total number of parameters for the embedding part is $|E_n|d + L|E_e|d$. The parameters in the message passing layer consist of the 2-layer MLP including biases shown in Eq. (2) and its subsequent batch normalization, with each layer having $L(2dd_1 + d + d_1 + 2d)$ parameters. In summary, the total number of parameters to update in full fine-tuning is

$$N_{Fine-Tuning} = |E_n|d + L(|E_e|d + 2dd_1 + 3d + d_1). \tag{24}$$

In our `Pin-Tuning` method, the parameters of the embedding layers are still updated. However, in each message passing layer, the original parameters are completely frozen, and the parts that require updating are the two feed-forward layers and the layer normalization in the bottleneck adapter

Table 4: Comparison of total model size. $^*$ indicates that the parameters are frozen.

|  | GS-Meta | Ours |
|---|---|---|
| Size of Molecular Encoder | 1.86M | 1.86M$^*$ |
| Size of Adapter | - | 0.21M |
| Size of Context Encoder | 0.62M | 0.62M |
| Size of Classifier | 0.18M | 0.27M |
| Size of Total Model | 2.66M | 2.96M |
| Size of Tunable Part of the Model | 2.66M | 1.10M |

Table 5: Dataset statistics.

| Dataset | Tox21 | SIDER | MUV | ToxCast | PCBA |
|---|---|---|---|---|---|
| #Compound | 7831 | 1427 | 93127 | 8575 | 437929 |
| #Property | 12 | 27 | 17 | 617 | 128 |
| #Train Property | 9 | 21 | 12 | 451 | 118 |
| #Test Property | 3 | 6 | 5 | 158 | 10 |
| %Positive Label | 6.24 | 56.76 | 0.31 | 12.60 | 0.84 |
| %Negative Label | 76.71 | 43.24 | 15.76 | 72.43 | 59.84 |
| %Unknown Label | 17.05 | 0 | 84.21 | 14.97 | 39.32 |

module, amounting to $L(2dd_2 + d + d_2 + 2d)$ parameters for this part. Therefore, the total number of parameters that need to be updated in our `Pin-Tuning` is

$$N_{Pin-Tuning} = |E_n|d + L(|E_e|d + 2dd_2 + 3d + d_2). \tag{25}$$

The difference in the number of parameters updated between the two tuning methods is $\Delta N = (d_1 - d_2)L(2d + 1)$.

## C.2 Total model size

We provide a comparison of total model size between our `Pin-Tuning` and the state-of-the-art baseline method, GS-Meta. The total model size consists of both frozen parameters and trainable parameters. The results are presented in Table 4. The total size of our model is comparable to GS-Meta, but the number of parameters that need to be trained is far less than GS-Meta.

## D Details of datasets

We carry out experiments in MoleculeNet benchmark [61] on five widely used few-shot molecular property prediction datasets:

- **Tox21**: This dataset covers qualitative toxicity measurements and was utilized in the 2014 Tox21 Data Challenge.
- **SIDER**: The Side Effect Resource (SIDER) functions as a repository for marketed drugs and adverse drug reactions (ADR), categorized into 27 system organ classes.
- **MUV**: The Maximum Unbiased Validation (MUV) is determined through the application of a refined nearest neighbor analysis, specifically designed for validating virtual screening techniques.
- **ToxCast**: This dataset comprises a compilation of compounds with associated toxicity labels derived from high-throughput screening.
- **PCBA**: PubChem BioAssay (PCBA) represents a database containing the biological activities of small molecules generated through high-throughput screening.

Dataset statistics are summarized in Table 5 and Table 6.

## E Details of baselines

We compare our `Pin-Tuning` with two types of baseline models for few-shot molecular property prediction tasks, categorized according to the training strategy of molecular encoders: trained-from-scratch methods and pre-trained methods.

Table 6: Statistics of sub-datasets of ToxCast.

| Assay Provider | #Compound | #Property | #Train Property | #Test Property | %Label active | %Label inactive | %Missing Label |
|---|---|---|---|---|---|---|---|
| APR | 1039 | 43 | 33 | 10 | 10.30 | 61.61 | 28.09 |
| ATG | 3423 | 146 | 106 | 40 | 5.92 | 93.92 | 0.16 |
| BSK | 1445 | 115 | 84 | 31 | 17.71 | 82.29 | 0.00 |
| CEETOX | 508 | 14 | 10 | 4 | 22.26 | 76.38 | 1.36 |
| CLD | 305 | 19 | 14 | 5 | 30.72 | 68.30 | 0.98 |
| NVS | 2130 | 139 | 100 | 39 | 3.21 | 4.52 | 92.27 |
| OT | 1782 | 15 | 11 | 4 | 9.78 | 87.78 | 2.44 |
| TOX21 | 8241 | 100 | 80 | 20 | 5.39 | 86.26 | 8.35 |
| Tanguay | 1039 | 18 | 13 | 5 | 8.05 | 90.84 | 1.11 |

*Trained-from-scratch methods:*

- **Siamese** [25]: Siamese is used to rank similarity between input molecule pairs with a dual network.

- **ProtoNet** [46]: ProtoNet learns a metric space for few-shot classification, enabling classification by calculating the distances between each query molecule and the prototype representations of each class.

- **MAML** [10]: MAML adapts the meta-learned parameters to achieve good generalization performance on new tasks with a small amount of training data and gradient steps.

- **TPN** [35]: TPN classifies the entire test set at once by learning to propagate labels from labeled instances to unlabeled test instances using a graph construction module that exploits the manifold structure in the data.

- **EGNN** [22]: EGNN predicts edge labels on a graph constructed from input samples to explicitly capture intra-cluster similarity and inter-cluster dissimilarity.

- **IterRefLSTM** [1]: IterRefLSTM adapts Matching Networks [53] to handle molecular property prediction tasks.

*Pre-trained methods:*

- **Pre-GNN** [20]: Pre-GNN is a classic pre-trained molecular model, taking the GIN as backbone and pre-training it with different self-supervised tasks.

- **Meta-MGNN** [14]: Meta-MGNN leverages Pre-GNN for learning molecular representations and incorporates meta-learning and self-supervised learning techniques.

- **PAR** [58]: PAR uses class prototypes to update input representations and designs label propagation for similar inputs in the relational graph, thus enabling the transformation of generic molecular embeddings into property-aware spaces.

- **GS-Meta** [73]: GS-Meta constructs a Molecule-Property relation graph (MPG) and redefines episodes in meta-learning as subgraphs of the MPG.

Following prior work [58], for the methods we reproduced, we use GIN as the graph-based molecular encoder to extract molecular representations. Specifically, we use the GIN provided by Pre-GNN [20] which consists of 5 GIN layers with 300-dimensional hidden units. Pre-GNN, Meta-MGNN, PAR, and GS-Meta further use the pre-trained GIN which is also provided by Pre-GNN.

# F  Implementation details

## F.1  Hardware and software

Our experiments are conducted on Linux servers equipped with an AMD CPU EPYC 7742 (256) @ 2.250GHz, 256GB RAM and NVIDIA 3090 GPUs. Our model is implemented in PyTorch version 1.12.1, PyTorch Geometric version 2.3.1 (https://pyg.org/) with CUDA version 11.3, RDKit version 2023.3.3 and Python 3.9.18. Our code is available at: `https://github.com/CRIPAC-DIG/Pin-Tuning`.

## F.2  Model configuration

For featurization of molecules, we use atomic number and chirality tag as original atom features, as well as bond type and bond direction as bond features, which is in line with most molecular

Table 7: 10-shot performance on each sub-dataset of ToxCast.

| Model | APR | ATG | BSK | CEETOX | CLD | NVS | OT | TOX21 | Tanguay |
|-------|-----|-----|-----|--------|-----|-----|----|-------|---------|
| ProtoNet | 73.58 | 59.26 | 70.15 | 66.12 | 78.12 | 65.85 | 64.90 | 68.26 | 73.61 |
| MAML | 72.66 | 62.09 | 66.42 | 64.08 | 74.57 | 66.56 | 64.07 | 68.04 | 77.12 |
| TPN | 74.53 | 60.74 | 65.19 | 66.63 | 75.22 | 63.20 | 64.63 | 73.30 | 81.75 |
| EGNN | 80.33 | 66.17 | 73.43 | 66.51 | 78.85 | 71.05 | 68.21 | 76.40 | 85.23 |
| Pre-GNN | 80.61 | 67.59 | 76.65 | 66.52 | 78.88 | 75.09 | 70.52 | 77.92 | 83.05 |
| Meta-MGNN | 81.47 | 69.20 | 78.97 | 66.57 | 78.30 | 79.60 | 69.55 | 78.77 | 83.98 |
| PAR | 86.09 | 72.72 | 82.45 | 72.12 | 83.43 | 74.94 | 71.96 | 82.81 | 88.20 |
| GS-Meta | _90.15_ | _82.54_ | _88.21_ | _74.19_ | _86.34_ | _76.29_ | _74.47_ | _90.63_ | _91.47_ |
| Pin-Tuning | **92.78** | **83.58** | **89.49** | **75.96** | **87.70** | **76.33** | **75.56** | **90.80** | **92.25** |
| $\Delta$*Improve.* | 2.92% | 1.26% | 1.45% | 2.39% | 1.58% | 0.05% | 1.46% | 0.19% | 0.85% |

Table 8: 5-shot performance on each sub-dataset of ToxCast.

| Model | APR | ATG | BSK | CEETOX | CLD | NVS | OT | TOX21 | Tanguay |
|-------|-----|-----|-----|--------|-----|-----|----|-------|---------|
| ProtoNet | 70.38 | 58.11 | 63.96 | 63.41 | 76.70 | 62.27 | 64.52 | 65.99 | 70.98 |
| MAML | 68.88 | 60.01 | 67.05 | 62.42 | 73.32 | 69.18 | 64.56 | 66.73 | 75.88 |
| TPN | 70.76 | 57.92 | 63.41 | 64.73 | 70.44 | 61.36 | 61.99 | 66.49 | 77.27 |
| EGNN | 74.06 | 60.56 | 64.60 | 63.20 | 71.44 | 62.62 | 66.70 | 65.33 | 75.69 |
| Pre-GNN | 80.38 | 66.96 | 75.64 | 64.88 | 78.03 | 74.08 | 70.42 | 75.74 | 82.73 |
| Meta-MGNN | 81.22 | 69.90 | 79.67 | 65.78 | 77.53 | 73.99 | 69.20 | 76.25 | 83.76 |
| PAR | 83.76 | 70.24 | 80.82 | 69.51 | 81.32 | 70.60 | 71.31 | 79.71 | 84.71 |
| GS-Meta | _89.36_ | _81.92_ | _86.12_ | _74.48_ | _83.10_ | _74.72_ | _73.26_ | _89.71_ | _91.15_ |
| Pin-Tuning | **89.94** | **82.37** | **87.61** | **75.20** | **85.07** | **75.49** | **74.70** | **90.89** | **92.14** |
| $\Delta$*Improve.* | 0.65% | 0.55% | 1.73% | 0.97% | 2.37% | 1.03% | 1.97% | 1.32% | 1.09% |

pre-training methods. Following previous works, we set $d = 300$. For MLPs in Eq. (2), we use the ReLU activation with $d_1 = 600$. Pre-trained GIN model provided by Pre-GNN [20] is adopted as the PTME in our framework. We tune the weight of update constraint (i.e., $\lambda$) in {0.01, 0.1, 1, 10}, tune the learning rate of inner loop (i.e., $\alpha_{\text{inner}}$) in {1e-3, 5e-3, 1e-2,5e-2,1e-1, 5e-1, 1, 5}, and tune the learning rate of outer loop (i.e., $\alpha_{\text{outer}}$) in {1e-5, 1e-4, 1e-3,1e-2,1e-1}. Based on the results of hyper-parameter tuning, we adopt $\alpha_{\text{inner}} = 0.5, \alpha_{\text{outer}} = 1e - 3$ and $d_2 = 50$. The `ContextEncoder`($\cdot$) described in Section 4.2 is implemented using a 2-layer message passing neural network [11]. In each MPNN layer, we employ a linear layer to aggregate messages from the neighborhoods of nodes and utilize distinct edge features to differentiate between various edge types in the context graphs. For baselines, we follow their recommended settings.

# G More experimental results and discussions

**More discussion of Figure 1.** The results show that molecular encoders with more molecule-specific inductive biases, such as CMPNN [47] and Graphormer [66], performed slightly worse than GIN-Mol [20] on this few-shot task. This is because more complex encoders require more parameters to provide inductive biases, which are difficult to train effectively under a few-shot setting.

**More main results.** The detailed comparison between `Pin-Tuning` and baseline models on sub-datasets of ToxCast are summarized in Table 7 and Table 8. Our method outperforms all baseline models under both the 10-shot and 5-shot settings, demonstrating the superiority of our method compared to existing methods.

**More discussion of ablation study.** Different datasets show varying sensitivity to the removal of components. On small-scale datasets like Tox21 and SIDER, removing components leads to a significant performance drop. On large-scale datasets like ToxCast and PCBA, the impact of removing components is less pronounced. This is because more episodes can be constructed on large-scale datasets, which aids in adaptation. This observation indicates that `Pin-Tuning` can bring considerable benefits in situations where data is extremely scarce.

**More case studies.** We provide more case studies in Figure 8 and 9 as a supplement to Section 5.5.

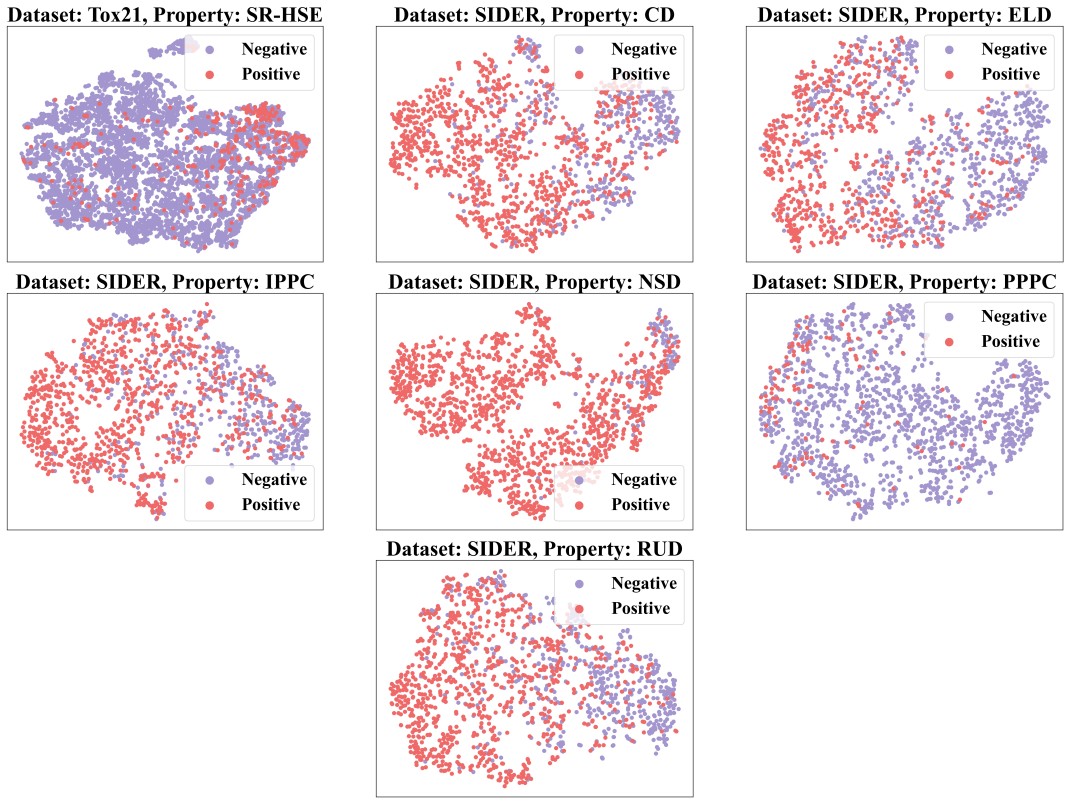

Figure 8: More molecular representations encoded by GS-Meta [73].

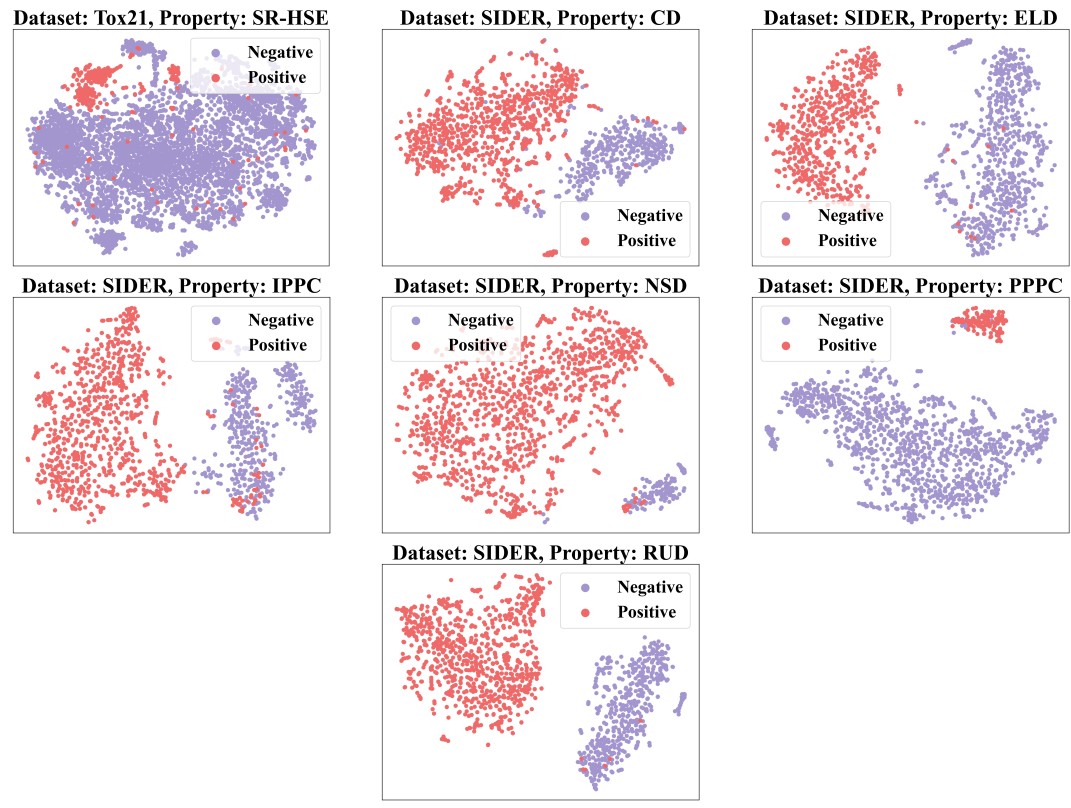

Figure 9: More molecular representations encoded by Pin-Tuning.

## H  Limitations and future directions

As we discusses in Section 5.2, although our method significantly outperforms the state-of-the-art baseline method, our method exhibits higher standard deviations in the experimental results under multiple runs with different seeds.

We further speculate that these high standard deviations might be due to the uncertainty in the context information within episodes. The explicitly introduced molecular context, on one hand, provides effective guidance for tuning pre-trained molecular encoders, but on the other hand, this information also carries a high degree of uncertainty. We aim to model the target property through the molecule-property relationships within episodes, but each episode is obtained by sampling very few samples from the large space corresponding to the target property. The uncertainty between different episodes is relatively high. How to quantify and calibrate this uncertainty is another question worth exploring, which we will investigate in our future work.

