# OpenReview forum: "Pin-Tuning: Parameter-Efficient In-Context Tuning for Few-Shot Molecular Property Prediction"
_NeurIPS.cc/2024/Conference — NeurIPS 2024 poster_

### Official Review · Reviewer_nYSV · 2024-07-10

**Soundness:** 2
**Presentation:** 3
**Contribution:** 2
**Rating:** 4
**Confidence:** 3

**Summary:**

This work introduce a new system for the FSMPP tasks, which includes: context adapter in GNN, context graph, and  weight consolidation. With these three techniques, the Pin-Tuning method achieved promising results on many FSMPP benchmarks.

**Strengths:**

- FSMPP tasks are important and have grate research value.
- The Pin-Tuning system is carefully designed and evaluated.

**Weaknesses:**

- The adapter based parameter efficient tunning have already been used in many tasks.

**Questions:**

- Why different strategies are used for embedding layers and encoder layers? What will be the results for (a) finetune encoder layers with weight constraint, and (b) freeze embedding layers and add adapters to them?

-  According to Table 3, the simplest weight consolidation method (i.e., IM) is the best. In this case, does the BWC theory really required as it is just L2 penalty on the changed parameters?

**Limitations:**

yes

---

> ### Author Rebuttal · Authors · 2024-08-07
>
> # Responses to Reviewer nYSV
>
> We thank the reviewer for your constructive feedback. Please find detailed responses below.
>
> > `W1` The novelty of our MP-Adapter for parameter efficient tunning.
>
> Based on your valuable feedback, we have provided a detailed description in the `Global Response` to clarify the specific design and considerations of our MP-Adapter for molecular representation fine-tuning. Additionally, we conducted experiments to empirically demonstrate the advantage of our method. **We kindly suggest referring to our `Global Response` regarding this issue.**
>
> > `Q1` The reasons for adopting different strategies for embedding layers and message passing layers, as well as the results of the opposite strategies.
>
> Our goal is to perform parameter-efficient tuning on pre-trained molecular encoders to address the imbalance between the abundance of tunable parameters and the scarcity of labeled molecules.
>
> - **For message passing layers, the number of parameters is disproportionately large compared to the training samples.** To mitigate this imbalance, we design a lightweight adapter targeted at the message passing layers, called MP-Adapter.
> - **Unlike message-passing layers, embedding layers have a very small number of parameters.** Therefore, we treat embedding layers in a different way than message passing layers by not using adapters. Instead, we directly fine-tune the parameters of the embedding layers, but impose a constraint called Emb-BWC to limit the magnitude of parameter updates, preventing aggressive optimization and catastrophic forgetting.
>
> Based on your constructive suggestions, we conducted experiments on the opposite design to further verify that the two strategies are suitable for their respective target components. Specifically, on the basis of our Pin-Tuning, we (Variant 1) changed the tuning strategy for the embedding layers to MP-Adapter, and (Variant 2) changed the tuning strategy for the message passing layers to Emb-BWC constraint.
>
> | Model | Tuning Strategy for Embedding Layers | Tuning Strategy for Message Passing Layers | Tox21 |  | SIDER |  | MUV | | PCBA |  |
> |-|-|-|-|-| - | - | - | - | - | - |
> | | |  | 10-shot | 5-shot | 10-shot | 5-shot | 10-shot | 5-shot | 10-shot | 5-shot |
> | Pin-Tuning (Ours) | Emb-BWC | MP-Adapter| 91.56   | 90.95  | 93.41   | 92.02  | 73.33   | 70.71  | 81.26   | 79.23  |
> | Variant 1   | MP-Adapter   | MP-Adapter   | 91.05| 89.19| 90.38 | 89.81  | 72.99   | 70.50  | 80.93   | 79.79  |
> |Variant 2| Emb-BWC|Emb-BWC|88.53|87.24|88.45|87.02|69.76|68.43|79.39|78.11|
>
> **Changing the tuning strategy of any component to the opposite on the basis of Pin-Tuning results in degraded performance**, which indicates the suitability of the two tuning strategies for their respective target components. Furthermore, **selecting the appropriate tuning strategy for the message passing layers has a greater impact on performance**. Particularly on the MUV dataset, adapter-based parameter-efficient fine-tuning of the message passing layers is key to achieving success on this dataset. For the reasons behind the degraded performance of the variants, we provide the following insights:
>
> - `Reasons why the Emb-BWC is not suitable for message passing layers`: First, the parameters of message passing layers are numerous and diverse in type. Therefore, applying the same weight constraint to numerous and diverse parameters cannot account for the differences in importance between parameters, making it less flexible than adapters. Second, in our method, adapters serve as carriers for introducing molecular context during fine-tuning, which cannot be achieved by directly updating the parameters. **Overall, the Emb-BWC constraint is not flexible enough for tuning modules with a large number of parameters and cannot introduce additional molecular context information.**
> - `Reasons why the MP-Adapter is not suitable for embedding layers`: The embedding layers function as lookup tables that map raw features into vectorized features, and they inherently have a small number of parameters. The advantage of adapters lies in their ability to perform parameter-efficient tuning. However, for the embedding layers, the number of parameters to be tuned is comparable whether directly updating the embedding layers or updating them through an adapter. **Overall, adapters are more suitable for modules with a large number of parameters, rather than for modules like the embedding layers with relatively few parameters.**
>
> > `Q2` Given that the simplest weight consolidation method (i.e., IM) is the best, does the BWC theory really required as it is just L2 penalty on the changed parameters?
>
> Although the simplest identity matrix approximation of Emb-BWC achieved the best results, we also obtained the following important observations from this set of experiments. **These observations reveal the intrinsic nature of the importance of parameters in pre-training and fine-tuning in molecular property prediction tasks, and can provide guidance on how to effectively fine-tune these parameters.**
>
> - Since the empirical results with three types of Emb-BWC regularizers are better than those without any regularizer, it indicates that imposing importance constraints on the parameters is beneficial for fine-tuning molecular representations.
> - The principle of the BWC theory is to measure the importance of parameters during pre-training and use this importance to constrain the fine-tuning process. This approach helps in preserving the knowledge acquired during pre-training by preventing significant changes to the important parameters during fine-tuning. It is observed that the more relaxed the Emb-BWC constraint, the better the fine-tuning performance. The explanation is that the important parameters in pre-training and the parameters that need to be retained during fine-tuning do not completely overlap, and some mechanisms of message passing require considerable updates during the fine-tuning process.

---

> > ### Author Response · Authors · 2024-08-14
> > **Thank you & Looking forward to your reply!**
> >
> > Dear reviewer nYSV,
> >
> > Thank you so much for your comments!
> >
> > We have provided a detailed clarification on the novelty of our MP-Adapter and conducted additional experiments on the tuning strategies for embedding layers and message passing layers. As the discussion period is drawing to a close with only a few hours remaining, we eagerly anticipate your response, which holds great significance for us.
> >
> > Best regards,
> >
> > Authors

---

> > ### Author Response · Authors · 2024-08-14
> > **Thank you & Looking forward to your reply!**
> >
> > Dear reviewer nYSV,
> >
> > Thank you so much for your comments. As the discussion period is drawing to a close, we eagerly anticipate your response, which holds great significance for us. Could we kindly know if our rebuttal has addressed your concerns?
> >
> > Best regards,
> >
> > Authors

---

> ### Author Response · Authors · 2024-08-07
> **Thank you & Looking forward to your reply**
>
> Thank you very much for your precious time and valuable comments. We hope our responses have addressed your concerns. Please let us know if you have any further questions. We are happy to discuss them further. Thank you.
>
> Best regards,
>
> Authors

---

### Official Review · Reviewer_u7hY · 2024-07-10

**Soundness:** 4
**Presentation:** 4
**Contribution:** 3
**Rating:** 7
**Confidence:** 4

**Summary:**

This paper proposes Pin-Tuning, a parameter-efficient in-context tuning method for few-shot molecular property prediction (FSMPP), mitigating the parameter-data imbalance and enhancing the contextual perceptiveness of pre-trained molecular encoders. This method treats the embedding layers and message passing layers in the pre-trained molecular encoder separately, based on specific reasons why they could not be effectively fine-tuned before. This includes a series of Bayesian weight consolidation constraints for the embedding layers and the bottleneck adapters for the message passing layers. Additionally, molecular context is introduced into the adapter modules to enable contextual perceptiveness.  Experiments show that the proposed method effectively improves few-shot performances across a range of benchmark datasets.

**Strengths:**

1. The motivation is convincingly derived from observations in the pilot experiment in Figure 1, indicating the necessity to design a more effective tuning method for few-shot molecular tasks.
2. The authors design different tuning methods for the embedding layer and the message passing layer in the pre-trained molecular encoder to accommodate their different parameter scales and parameter forms.
3. The proposed Emb-EWC constraint is derived based on Bayesian learning theory, and the authors provide intuitive explanations for each approximation choice to make them easier to understand.

**Weaknesses:**

1. It is recommended to add details about the pilot experiment in Figure 1, such as the featurization and the specific architectures of different models, and whether the pre-training strategies are consistent.
2. Table 4 only reports the "total size of the molecular encoder." Considering that some parameters are frozen in your method, you could add a row for the "size of the part that needs to be tuned". This would be helpful to make the advantages of your method clearer.

**Questions:**

1. What are the details of the pilot experiment in Figure 1? Including the featurization, the model architecture, and the pre-training strategy of each model. More details are needed to assess the fairness of the experiment and to strengthen the persuasiveness of your claims.
2. Does $\mathcal{L}_{update}$ in Figure 2(d) refer to $\mathcal{L}_{Emb-BWC}$ in Section 4.1.2? The content in the figure should be consistent with the text.

**Limitations:**

Yes.

---

> ### Author Rebuttal · Authors · 2024-08-07
>
> # Responses to Reviewer u7hY
>
> We thank the reviewer for your constructive feedback. Please find detailed responses below.
>
> > `W1` `Q1` Add details on pilot experiment in Figure 1.
>
> Thank you very much for your reply. We apologize for the lack of details about the pilot experiment due to the length limit of the paper. We further supplement the details of the comparison of GIN-Mol [1], CMPNN [2], and Graphormer [3] as follows.
>
> - `Featurization`: To ensure a fair comparison with existing methods for few-shot molecular property prediction (FSMPP), we employed the same featurization approach as them. Following prior works of molecular representation learning [1] and few-shot molecular property prediction [4], we used two atomic features (`atomic number` and `chiral tag`) and two bond features (`bond type` and `bond direction`).
> - `Architecture Choice`: GIN-Mol is a GIN tailored to molecules, proposed in Pre-GNN [1]. This molecular GIN has become a representative encoder in molecular pretraining, which is equipped with multiple bond embedding layers and introduces bond features into message passing. **The standard 5-layer Mol-GIN is adopted**. To facilitate interaction between node and directed edge information, CMPNN introduces a node-edge message communication module and offers multiple architectural options. We selected the **multilayer perception (MLP) to implement the message communication module**, since experiments with CMPNN have demonstrated that this architecture performs best. For Graphormer, we adopted the standard architecture based on multi-head attention, **using node degrees, shortest distances between nodes, and edge types to construct the position encoding**.
> - `Hyperparameters`: We adopted the hyperparameters provided by Mol-GIN, CMPNN and Graphormer. Primarily, the size of the hidden dimension is 300 and the number of encoding layers is 5. For Graphormer, the number of heads is set to 8, and the dropout rate for attention is 0.1.
> - `Pre-training and Adaptation Strategy`: To ensure a fair comparison, we followed prior FSMPP methods that use a pre-trained GIN as the molecular encoder. We pre-trained GIN-Mol, CMPNN, and Graphormer **with both the supervised task and the `Context Prediction` self-supervised task** [1], then adapted them to the FSMPP task through fine-tuning.
>
> > `W2` Make a comparison in terms of tunable parameter size in Table 4.
>
> Thank you for your valuable feedback. We appreciate your suggestion to provide additional details in Table 4 regarding the size of the tunable part of the model. In response to your comment, we have added a new row to Table 4 that specifies the size of the parameters tuned in our method, thereby better illustrating the advantages of our approach. The revised Table 4 is shown below, and is also provided in the `Global Response` PDF.
>
> || GS-Meta | Ours|
> | - | - | - |
> | Size of Molecular Encoder   | 1.86M | 1.86M  |
> | Size of MP-Adapter       | -  | 0.21M |
> | Size of Context Encoder   | 0.62M | 0.62M |
> | Size of Classifier    | 0.18M | 0.27M |
> | Size of Total Model   | 2.66M | 2.96M |
> | **Size of Tunable Part of the Model** | **2.66M** | **1.10M** |
>
> > `Q2` Inconsistency in the notion of the Emb--BWC regularizer between Figure 2 and Section 4.1.2.
>
> Yes, $\mathcal{L}\_{update}$ in Figure 2 refers to $\mathcal{L}\_{Emb-BWC}$. Thank you for pointing out this mistake and helping us improve the clarity of the paper. We have revised this figure in the Global Response PDF.
>
> **References**
>
> [1] Strategies for Pre-training Graph Neural Networks. ICLR 2020
>
> [2] Communicative Representation Learning on Attributed Molecular Graphs. IJCAI 2020
>
> [3] Do Transformers Really Perform Bad for Graph Representation? NeurIPS 2021
>
> [4] Property-Aware Relation Networks for Few-Shot Molecular Property Prediction. NeurIPS 2021

---

> ### Author Response · Authors · 2024-08-07
> **Thank you & Looking forward to your reply**
>
> Thank you very much for your precious time and valuable comments. We hope our responses have addressed your concerns. Please let us know if you have any further questions. We are happy to discuss them further. Thank you.
>
> Best regards,
>
> Authors

---

> > ### Comment · Reviewer_u7hY · 2024-08-14
> >
> > Thanks for your efforts in the rebuttal. The rebuttal addressed my concerns. The additional details regarding the pilot experiment demonstrate that the comparison is fair and reasonable, making the motivation more convincing. Moreover, the revised table and figure improve the clarity of the paper. I have also read the comments from other reviewers and all discussions, and I think this paper presents a promising approach for the community. Therefore, I have decided to raise my score and lean towards acceptance.

---

> > > ### Author Response · Authors · 2024-08-14
> > > **Thank you!**
> > >
> > > Thank you very much for your response! We are very glad that we have addressed your concerns. If you have any further questions, feel free to ask and we would be more than delighted to answer.
> > >
> > > Best regards,
> > >
> > > Authors

---

### Official Review · Reviewer_W6tN · 2024-07-11

**Soundness:** 2
**Presentation:** 2
**Contribution:** 2
**Rating:** 5
**Confidence:** 4

**Summary:**

The authors propose a strategy for few-shot drug discovery scenarios. The authors propose to fine-tune representations retrieved from encoder layers with adapters. In addition to the initial representations from the message passing layers, the adapters are provided with property representations and "context-aware" molecule representations to improve the initial representations. Both the property representations and the "context-aware" molecule representations stem from a context encoder. The context encoder creates learned molecule and property representations by relating the input molecule representations and the current few shot task, i.e. the property with other molecules and other already seen properties. The authors evaluate their approach in several datasets and include an ablation study.

**Strengths:**

**Originality**:
- **(S-O)**:  The Context encoder contains novelty. Jointly learning task and molecule representations with GNNs and updating molecule representation based on relationships between them is novel (also see (S-Q2)).

**Quality**:
- **(S-Q1)**: The model shows strong performance for the chosen experimental setup. In the main experiment (Table 1) the authors include error bars and therefore follow best practices of comparing drug discovery models.
- **(S-Q2)**: With their novel strategy to include in-context information, i.e. task information, the authors found a way to use label information of query and support set molecules present in the training set. This seems promising and has not been done before (S-O). E.g., compare [1] in which molecule representations are updated being aware of training molecules but not-aware of their training labels.

**Clarity**:
- **(S-C): Chapter 4 introduces very well the goal of this work. Taking into account all three textual descriptions formulas and Figure 2, the proposed method and its components become clear.

**Significance**:
- **(S-S1)**: The novel Context Encoder is relevant for the community (see (S-Q2)).
- **(S-S2)**: For the chosen experimental setup, the author's approach outperforms chosen baseline methods.

[1] Schimunek, Johannes, et al. "Context-enriched molecule representations improve few-shot drug discovery." The Eleventh International Conference on Learning Representations.

**Weaknesses:**

**Originality**:
- **(W-O1)**: Using adapters to efficiently fine-tune models is not novel. The core idea is to efficiently fine-tune a layer by adding another learnable component with less parameters. Instead of changing the model layer itself, the new component learns to modify the outcome of the model layer. This has already been done in few-shot drug discovery [2].
- **(W-O2)**: In-context fine-tuning in the sense of adapting molecular representations to the few-shot task is not novel and has been done e.g. here [1, 3] .

**Quality**:
- **(W-Q1)**: The authors missed to include the SOTA benchmark for few-shot drug discovery FS-Mol.
- **(W-Q2)**: The authors missed that in-context fine-tuning (W-O2) and efficient adapter-like fine-tuning (W-O1) has already been applied to few-shot drug discovery.
 - **(W-Q3)**: Missing error bars for Figure 1, Table 2, Figure 4, and Figure 5

**Significance**:
- **(W-S1)**: The significance of the experimental section is diminished because the SOTA benchmark for few-shot drug discovery is missing (see (W-Q1))
- **(W-S2)**: The significance of the proposed method is diminished because of (W-Q2). Being aware of the ideas which have already been applied to few-shot drug discovery would have allowed to put the context encoder into the focus of this work. The ablation study indicates that this module is important. Still in the current version of the paper, this novel piece of architecture remains underexplored.

**Clarity**:
- **(W-C1)**: Because of (W-S2, W-Q2) it remains unclear which parts of the proposed method have been tested in related work and which parts are novel. The authors might think about changing the main story line of their paper by focusing on their training label aware-fine-tuning procedure.
- **(W-C2)**: The procedure which happens inside the context encoder is not displayed well in Figure 2. Since this is a central part, this should be improved.
- **(W-C3)**: English editing is required
- **(W-C4)**: The mathematical notation and formulas seem cluttered and inconsistent. E.g., in 3.2. a molecular encoder is denoted by $f(.)$ but the context encoder is denoted by ContextEncoder. Generally, the authors should stick to standard notation and avoid "full-word notation" in formulas.


[2] Adler, Thomas, et al. "Cross-domain few-shot learning by representation fusion." (2020).\
[3] Altae-Tran, Han, et al. "Low data drug discovery with one-shot learning." ACS central science 3.4 (2017): 283-293.

**Questions:**

-

**Limitations:**

Limitations have been addressed in the appendix.

---

> ### Author Rebuttal · Authors · 2024-08-07
>
> # Responses to Reviewer W6tN (1/3)
>
> We thank the reviewer for your constructive feedback. Please find detailed responses below.
>
> > `W-O1` `W-Q2` The novelty of our adapter-based parameter-efficient tuning.
>
> We greatly appreciate your comments. Indeed, there are several adapter-like methods used to fine-tune pre-trained models in various fields. Unlike other adapters, our MP-Adapter is specifically designed for molecular encoders based on graph neural networks (GNNs) that follow the message passing mechanism. Here, **we compare our MP-Adapter with two potentially similar methods to highlight the novelty of our approach**: (1) adapters used in natural language processing (NLP) for transformer-based language models, and (2) the representation fuser CHEF proposed in previous few-shot drug discovery work [1] designed for fully-connected deep neural networks (FCNs).
>
> 1. `The comparison between our MP-Adapter and adapters used in NLP`: This comparison has been provided in our `Global Response` to clarify the specific design and considerations of our MP-Adapter for molecular representation fine-tuning. Additionally, we conducted experiments to compare the performance of fine-tuning molecular encoders using NLP adapters and our MP-Adapter, empirically demonstrating the advantages of our method. **We kindly suggest referring to our `Global Response` regarding this issue.**
>
> 2. `The comparison between our MP-Adapter and the representation fuser CHEF [1]`:  CHEF fuses representations from different layers by attaching a learnable learner after each frozen pre-trained layer and ensembling the outputs of all learners. A critical difference between CHEF and our MP-Adapter is that **in CHEF, the output of each learner is fed into the final fuser for ensembling rather than being fed to the next layer of the encoder**. Therefore, in the CHEF framework, updating the output of the previous layer through the added learner does not affect the output of subsequent layers. **This approach is only suitable for global encoding models, such as the FCNs used in the experiments evaluating CHEF, which take molecular fingerprints (ECFP6) as input [1].** However, the current state-of-the-art molecular encoders are based on GNNs and take molecular graphs as input. GNN-based molecular encoders follow a message passing mechanism to achieve localized neighborhood information aggregation. In this case, the interaction between different layers of the encoder is more important, and changes in the output of the previous layer will significantly affect the message passing of the next layer. Therefore, **passing the adapter's output to the next encoding layer as input, as our MP-Adapter does, is more appropriate for GNN-based molecular encoders.**
>
> Overall, neither NLP adapters nor the representation fuser CHEF can be trivially used to fine-tune GNN-based molecular encoders. We have also summarized the comparison of them in the table below to clearly illustrate the advantages and novelty of our MP-Adapter:
>
> |                | Target Model                      | Target Component                          | Suitable for Localized Encoding Layer | Capable of Perceiving Molecular Context |
> | -------------- | --------------------------------- | ----------------------------------------- | ------------------------------------- | --------------------------------------- |
> | CHEF [1]       | FCNs                              | Fully-connected Layer                     | No                                    | No                                      |
> | NLP Adapter    | Transformer-based Language Models | Multi-head Attention / Feed Forward Layer | Yes                                   | No                                      |
> | Our MP-Adapter | GNN-based Molecular Encoders      | Message Passing Layer                     | Yes                                   | Yes                                     |
>
> > `W-O2` `W-Q2` Novelty of in-context fine-tuning.
>
> Although some previous works have also mentioned the concept of `molecular context`, our work differs in the definition, encoding method, and perceiving method of molecular context.
>
> - `Definition of molecular context`: In IterRefLSTM [2] and PAR [3], the context refers to the structural and property labels of support molecules concerning the target property, while in MHNfs [4], the context refers to molecules that are structurally similar to the given molecule among a large number of unlabeled molecules. Unlike these methods, **the molecular context in our approach refers to the label information of seen and unseen molecules in auxiliary properties, which are informative and more relevant to the target task**.
> - `Encoding of molecular context`: Previous works [2,3,4] typically use LSTM or attention mechanisms to learn the correlations between molecules, often neglecting the interactive relationship between molecules and properties. **We represent the molecular context as a molecule-property bipartite graph and encode it using a GNN-based context encoder, which allows for more effective and robust learning of the relationships between molecules and properties.**
> - `Perceiving of molecular context`: Previous works [2,3,4] combine non-contextual molecular representations with context representations to predict molecular properties. In this paradigm, the resulting molecular representations cannot perceive the molecular context information. **In our method, the molecular context is introduced into the fine-tuning process through the MP-Adapter, allowing the molecular encoder to be fine-tuned under the guidance of the context, thereby obtaining contextual molecular representations.**

---

> ### Author Response · Authors · 2024-08-07
> **Rebuttal by Authors**
>
> # Responses to Reviewer W6tN (2/3)
>
> > `W-Q1` `W-S1` Experiments on the FS-Mol benchmark.
>
> Thank you for your constructive suggestions. We have added experimental results on the FS-Mol benchmark. Note that the data distribution of FS-Mol is significantly different from that of another import benchmark MoleculeNet:
>
> 1. FS-Mol only includes properties related to the biological activity of small molecules against protein targets, unlike MoleculeNet, which encompasses a variety of property types such as toxicity, solubility, blood-brain barrier penetration, and side effects.
> 2. FS-Mol contains over 5,000 tasks with 233,786 unique compounds, and 489,133 measurements have been conducted. Although FS-Mol covers a wide range of molecules and properties, the **number of molecules measured for each property is relatively small, and there is little overlap between the molecules covered by different properties**. Therefore, **it is difficult to provide effective context between different properties on the original FS-Mol dataset**.
>
> To support our claim regarding the sparse context in FS-Mol, we conducted a statistical analysis of the distribution in the original FS-Mol dataset:
>
> - The number of properties measured for each molecule
>
> | Max  | Min  | Mean | 25th percentile | 50th percentile | 75th percentile | 90th percentile | 95th percentile |
> | ---- | ---- | ---- | - | - | - | - | - |
> | 602  | 1    | 1.94 | 1   | 1   | 2   | 3   | 4   |
>
> - The number of molecules commonly measured between two properties
>
> | Max  | Min  | Mean | 25th percentile | 50th percentile | 75th percentile | 90th percentile | 99th percentile |
> | -- | ---- | ---- | - | - | - | - | - |
> | 2230 | 0    | 0.79 | 0  | 0  | 0 | 0 | 7 |
>
> Since the original FS-Mol is not suitable for encoding and perceiving molecule context based on labels, **we designed two evaluation settings**.
>
> `Evaluation Setting 1`: **We evaluated our parameter-efficient tuning method under the standard FS-Mol evaluation setting, without considering molecular context.** Based on PAR [3], which is the state-of-the-art baseline model irrelevant to the context from auxiliary tasks, we tested the results of introducing MP-Adapter and Emb-BWC. We tested different support set sizes of 16, 32, and 64. We run 10 times with different seeds and report the average ΔAUPRC.
>
> |   | FS-Mol (16-shot) | FS-Mol (32-shot) | FS-Mol (64-shot) |
> | - | - | - | - |
> | PAR | 47.94±0.23 | 48.18±0.21 | 48.73±0.37 |
> | PAR + MP-Adpater| 49.33±0.19 | 49.43±0.26 | 49.80±0.26 |
> | PAR + Emb-BWC | 48.14±0.24 | 49.80±0.33| 51.01±0.48 |
> | PAR + MP-Adapter + Emb-BWC | 49.76±0.16 | 49.96±0.13 | 50.14±0.27 |
>
> `Evaluation Setting 2`: **To better evaluate the effectiveness of our context encoding and in-context tuning on the FS-Mol dataset, we selected two subsets of data from FS-Mol with dense context information to construct two sub-datasets and evaluated them using the standard N-way K-Shot setting.** The FS-Mol-6K dataset contains 15 properties and over 6,000 molecules, while the FS-Mol-24K dataset contains 128 properties and over 24,000 molecules. Experiments were conducted under 10-shot and 5-shot settings with 10 different seeds, using the average AUC-ROC score as the evaluation metric. These two constructed sub-datasets have been uploaded to our anonymous code repository.
>
> | | FS-Mol-6k (10-shot) | FS-Mol-6k (5-shot) | FS-Mol-24K (10-shot) | FS-Mol-24K (5-shot) |
> | - | - | - | - | - |
> | PAR  | 78.52±0.33  | 78.19±0.23| 63.43±0.41  | 62.25±0.28 |
> | GS-Meta  | 80.36±0.73  | 79.58±0.80 | 67.28±0.66 | 66.86±0.32 |
> | Pin-Tuning (Ours) | 82.28±1.99 | 81.77±1.71 | 68.94±0.69| 68.02±0.93 |
>
> From the evaluation results of the two settings, **our parameter-efficient tuning method outperforms the baseline methods on the FS-Mol benchmark, regardless of whether molecular context is considered.**

---

> ### Author Response · Authors · 2024-08-07
> **Rebuttal by Authors**
>
> # Responses to Reviewer W6tN (3/3)
>
> > `W-S2` `W-C1` The contribution of parameter-efficient tuning and context encoding.
>
> In the above response, we have clarified the novelty of our parameter-efficient tuning and in-context tuning. Here, we further clarify their contributions to performance improvement. **Both parameter-efficient tuning and context encoding can contribute to performance improvement. Specifically, on the MUV dataset, fine-tuning the message passing layers with an adapter plays a decisive role, having a greater impact than context**, as observed from the experimental results in our `Global Response`. Therefore, the influence of parameter-efficient tuning and context encoding on performance improvement depends on the data distribution, and each may significantly contribute on different datasets.
>
> > `W-C2` `W-C3` `W-C4` Suggestions for improving the quality and clarity of the presentation.
>
> Thank you for your suggestions to improve the presentation of the paper. We have made the following efforts:
>
> 1. We have added a description of the context encoder in Figure 2. The revised Figure 2 is provided in the Global Response PDF.
> 2. We have performed language editing and proofreading to enhance the clarity and readability of the paper.
> 3. We have standardized the use of mathematical notation and formulas. We replaced notation like "$ContextEncoder(\cdot)$" with letter-based notation to make the usage more standard and consistent.
>
> **References**
>
> [1] Cross-Domain Few-Shot Learning by Representation Fusion. 2020
>
> [2] Low Data Drug Discovery with One-Shot Learning. ACS Central Science 2017
>
> [3] Property-Aware Relation Networks for Few-Shot Molecular Property Prediction. NeurIPS 2021
>
> [4] Context-Enriched Molecule Representations Improve Few-Shot Drug Discovery. ICLR 2023

---

> ### Author Response · Authors · 2024-08-07
> **Thank you & Looking forward to your reply**
>
> Thank you very much for your precious time and valuable comments. We hope our responses have addressed your concerns. Please let us know if you have any further questions. We are happy to discuss them further. Thank you.
>
> Best regards,
>
> Authors

---

> ### Comment · Reviewer_W6tN · 2024-08-12
> **Answer to the authors' rebuttal**
>
> ### Adapter / CHEF discussion:
> - GNNs / MLPs:
> > However, the current state-of-the-art molecular encoders are based on GNNs and take molecular graphs as input.
>
>     This is a minor point but still worth mentioning: I'd disagree here if it was meant in the way that GNNs are SOTA and MLPs are not. As far as I know, generally and merged across multiple bioactivity datasets, both can be considered SOTA. Also see [1] which combine GNNs with MLPs to reach SOTA performance.
>
> - CHEF for GNNs:
> > This approach is only suitable for global encoding models, such as the FCNs used in the experiments evaluating CHEF, which take molecular fingerprints (ECFP6) as input
>
>     I do think CHEF suits well for GNNs. The core idea of CHEF is to use both low-level and higher-level features for domain adaptation. This fits very well with GNNs, as representations retrieved from very early layers can be interpreted as very local features (i.e. a low-level feature), while representations from later layers capter larger parts of the molecular graph (i.e. a higher-level feature).
>
> - This is why CHEF can be used to fine-tune GNN-based molecular encoders
> > neither NLP adapters nor the representation fuser CHEF can be trivially used to fine-tune GNN-based molecular encoders.
> - Novelty:
> The way label information is included is novel. Also, I appreciate the  authors' reasoning that the model might benefit from feeding the adapter's output into the next layer. However, the authors missed to include experiments which show that their claim really holds.
>
> ### Context discussion:
> I appreciate the authors' discussion here and I mostly agree. (The MHNfs' representations could be considered as contextual molecular representations also because the enrichment step is analogous to a LLM which uses information from a large context-window to update token representations). The authors' main contribution is their context-guided fine-tuning, while context means both structural and label information. This is valuable to the community. However, both the quality of the manuscript as well as the value for the community heavily depend on an extensive discussion about similarities and differences in how different approaches use context. Reading answer 3/3, I am not convinced the authors addressed this enough in their current version of the manuscript.
>
> ### FS-Mol experiment:
> - An experiment with support set size of 16 is not a 16-shot experiment. A support set size of 16 means that the total number of available labeled samples for tasks during inference is 16, e.g. 6 actives and 10 inactives. For the FS-Mol benchmark experiment  the support set is created using a stratified split, compare [2].
> - The reported $\Delta$AUPRC values are very high. Recently published SOTA methods are typically reported with performance values between 0.2 and 0.3 (*) for the support set sizes 16 and 32. Considering that a random classifier achieves AUCPR values around 0.5, the reported AUCPR values indicate close-to-perfect classifiers. Given this significant performance gap, and considering that PAR has already been evaluated on FS-Mol with $\sim\Delta$AUPRC 0.17 [3], an explanation of why the evaluated models performed so well in the included experiment would be valuable.
>
> ### Summary:
> I appreciate the work the authors put into the rebuttal, I see the potential of this work, and I generally would evaluate this work to be interesting for the community. However, I also think this manuscript would benefit from another review round (see adapter discussion, context discussion, and FS-Mol discussion) since some improvements still seem crucial to eventually end with a manuscript of high quality. For this reason, I believe it is too early to publish this work at this conference, and I therefore stick to my initial rating.
>
> [1] Yang, Kevin, et al. "Analyzing learned molecular representations for property prediction." Journal of chemical information and modeling 59.8 (2019): 3370-3388.
>
> [2] Stanley, Megan, et al. "Fs-mol: A few-shot learning dataset of molecules." Thirty-fifth Conference on Neural Information Processing Systems Datasets and Benchmarks Track (Round 2). 2021.
>
> [3] Context-Enriched Molecule Representations Improve Few-Shot Drug Discovery. ICLR 2023

---

> > ### Author Response · Authors · 2024-08-14
> > **Response to the reviewer's comments (1/2)**
> >
> > Dear review W6tN,
> >
> > We sincerely appreciate your thoughtful response to our rebuttal and the valuable feedback you have provided. Your insights are incredibly helpful in enhancing the quality of our paper. We hope to address each of your remaining concerns point by point below.
> > ### Adapter / CHEF discussion:
> >
> > > Both GNNs and MLPs can be considered SOTA.
> >
> > Thank you very much for your insightful comments. We completely agree with your perspective. Both GNNs and MLPs are highly effective for encoding molecular data. They each have significant advantages when dealing with molecular data characterized by topological graphs and molecular fingerprints, respectively, and can both be considered SOTA molecular encoders.
> >
> > Additionally, the D-MPNN [1] you mentioned is indeed a very representative molecular encoder. In our paper, we discuss GIN-Mol [2] and CMPNN [3], which are simplified and enhanced versions of D-MPNN, respectively. The difference between them is that D-MPNN designs the edge-based message passing mechanism on the basis of GIN-Mol to prevent totters, while CMPNN strengthens the interaction between atoms and bonds based on D-MPNN. The commonality among these three models is that they all use the GNN backbone based on the Graph Isomorphism Network (GIN) [4], which is known for its high expressive power (generalizes the WL test). This high expressive power comes from the aggregation scheme based on MLPs, which is considered injective on multisets (see Theorem 3 and Corollary 6 in [4]). Therefore, we fully agree with your point that MLPs are SOTA models on molecular data, and incorporating MLPs as the aggregate function in GNNs endows GNNs with stronger expressive power.
> >
> > We also apologize for the inappropriate expression in our initial rebuttal. What we intended to convey is that for many molecular encoders with GNN backbones, our MP-Adapter suits their message passing mechanisms by feeding the output of the adapters into the next encoding layer.
> >
> > > CHEF suits well for GNNs and can be used to fine-tune GNN-based molecular encoders.
> >
> > We greatly appreciate your insightful perspective that CHEF [5] is capable of aggregating low-level features from the shallow layers and higher-level features from the deeper layers of GNNs. Considering the suitability of CHEF for fine-tuning GNNs and its distinction from adapters, we have added experiments to fine-tune the GNN-based encoder using CHEF.
> >
> > |                                   | Tox21   |        | SIDER   |        | MUV     |        | PCBA    |        |
> > | --------------------------------- | ------- | ------ | ------- | ------ | ------- | ------ | ------- | ------ |
> > |                                   | 10-shot | 5-shot | 10-shot | 5-shot | 10-shot | 5-shot | 10-shot | 5-shot |
> > | No Adapter                        | 86.67   | 86.43  | 84.36   | 84.57  | 66.08   | 64.50  | 79.40   | 77.47  |
> > | **CHEF**                          | 87.24   | 87.18  | 84.81   | 84.52  | 69.30   | 67.65  | 79.69   | 77.20  |
> > | NLP Adapter                       | 87.94   | 87.80  | 85.18   | 84.79  | 71.86   | 69.58  | 79.75   | 78.35  |
> > | MP-Adapter                        | 90.17   | 89.59  | 92.06   | 91.43  | 72.37   | 71.65  | 80.74   | 78.51  |
> > | Pin-Tuning (MP-Adapter + Emb-BWC) | 91.56   | 90.95  | 93.41   | 92.02  | 73.33   | 70.71  | 81.26   | 79.23  |
> >
> > On most datasets, CHEF performs comparably to standard NLP adapters, as both use lightweight trainable components to fine-tune the output of frozen pre-trained molecular encoder. Their performance lags behind that of our MP-Adapter because they do not integrate molecular context with the message passing process. We will include these experimental results and related discussions about CHEF in the paper.
> >
> > **References**
> >
> > [1] Analyzing Learned Molecular Representations for Property Prediction. Journal of Chemical Information and Modeling. 2019
> >
> > [2] Strategies for Pre-training Graph Neural Networks. ICLR. 2020
> >
> > [3] Communicative Representation Learning on Attributed Molecular Graphs. IJCAI. 2020
> >
> > [4] How powerful are graph neural networks? ICLR. 2019
> >
> > [5] Cross-Domain Few-Shot Learning by Representation Fusion. 2020

---

> > ### Author Response · Authors · 2024-08-14
> > **Response to the reviewer's comments (2/2)**
> >
> > ### Context discussion:
> >
> > > An extensive discussion about similarities and differences in how different approaches use context.
> >
> > Thank you for recognizing our work. We understand your concerns regarding the discussion on context modeling. It is indeed necessary to further expand the manuscript's discussion on this topic, providing a detailed analysis of the similarities and differences among different approaches, and clarifying the unique aspects and contributions of our approach. We will add the following independent paragraph to the related work section:
> >
> > "**Context modeling in few-shot molecular property prediction.** Recent efforts have shifted towards leveraging the unique nature of molecular property prediction, specifically the many-to-many relationships between molecules and properties that arise from the multi-labeled nature of molecules, often referred to as the *molecular context*. IterRefLSTM considers the structures and property labels of seen molecules with respect to the target property during prediction. PAR initially connects similar molecules for the target property using a homogeneous context graph. MHNfs introduces a large-scale external molecular library as context to augment the limited known information. However, the contexts constructed in these approaches are not informative enough, as they neglect the interactive relationship between molecules and properties. Unlike these methods, the molecular context in our approach includes the label information of both seen and unseen molecules in auxiliary properties. We represent the molecular context as a molecule-property bipartite graph and encode it using a GNN-based context encoder, which allows for effective and robust learning of the relationships between molecules and properties. Furthermore, the molecular context is introduced into the fine-tuning process through our MP-Adapter, enabling the molecular encoder to be fine-tuned under the guidance of the context, thereby obtaining contextual molecular representations."
> >
> > We hope this addition will enhance the quality of the manuscript and clarify its value to the community.
> >
> > ### FS-Mol experiment:
> >
> > > An experiment with support set size of 16 is not a 16-shot experiment.
> >
> > Thank you for pointing out this discrepancy regarding the support set size. We apologize for the confusion caused by our mention of "16-shot" in our rebuttal. This was indeed a typo. In our experiments, we actually compared different support set sizes based on stratified sampling. Specifically, we followed the settings of MHNfs [6] and FS-Mol [7], where we conducted meta-training with a support set size of 64 and stratified sampling. For evaluation, we used support set sizes of 16, 32, 64, 128, and 256, also based on stratified sampling.
> >
> > We have corrected this typo and fixed the previous mistakes in our experiments. Below, we provide the revised table with the correct experimental results.
> >
> > > The reported $\Delta$AUPRC values are very high.
> >
> > Thank you for pointing out this issue. Upon re-examining our previous experimental results, we realized that we reported AUPRC instead of $\Delta$AUPRC. Considering that the AUPRC of a random classifier on FS-Mol is around 0.46 (since FS-Mol is not completely class-balanced), the actual $\Delta$AUPRC of our previous results is approximately 0.03 to 0.05.
> >
> > Given that these are very poor results, we carefully reviewed our experimental process. We found that the reason for the poor results was that we mistakenly used a model checkpoint saved early in the training process for evaluation. The complete training process consists of 10,000 epochs, but the evaluated model was saved around the 200th epoch. We have corrected this mistake and evaluated the fully trained model.
> >
> > Using the correct model checkpoint and the correct $\Delta$AUPRC metric, we have re-evaluated our experimental results on the FS-Mol benchmark. Below, we provide the revised table with the correct experimental results:
> >
> > |Support Set Size|16|32|64|128|256|
> > |-|-|-|-|-|-|
> > |PAR|0.1578±0.0336|0.1669±0.0261|0.1723±0.0305|0.1917±0.0641|0.1561±0.0266|
> > |PAR + MP-Adpater|**0.1728±0.0375**|0.1815±0.0253|0.1833±0.0289|0.1984±0.0638|0.1894±0.0266|
> > |PAR + Emb-BWC|0.1620±0.0291|0.1691±0.0228|0.1743±0.0297|0.1905±0.0644|0.1699±0.0239|
> > |PAR + MP-Adapter + Emb-BWC|0.1721±0.0292|**0.1827±0.0216**|**0.1893±0.0281**|**0.2049±0.0603**|**0.2041±0.0283**|
> >
> > Combining the results in this table with the results from another setting in our rebuttal, our parameter-efficient tuning method demonstrates its effectiveness on the FS-Mol benchmark, regardless of whether molecular context is considered.
> >
> > ---
> >
> > Thank you once again for your constructive comments and for helping us improve the quality and clarity of our work. We sincerely hope that our response can address your concerns.
> >
> > **References**
> >
> > [6] Context-Enriched Molecule Representations Improve Few-Shot Drug Discovery. ICLR 2023
> >
> > [7] FS-Mol: A Few-Shot Learning Dataset of Molecules. NeurIPS 2021

---

> ### Comment · Reviewer_W6tN · 2024-08-14
> **Reviewer's response to 2nd rebuttal**
>
> > What we intended to convey is that for many molecular encoders with GNN backbones, our MP-Adapter suits their message passing mechanisms
>
> Understood!
>
> > we have added experiments to fine-tune the GNN-based encoder using CHEF. [...] Their performance lags behind that of our MP-Adapter because they do not integrate molecular context
>
> This improves the quality of the manuscript. The reasoning w.r.t. the molecular context makes sense.
>
> > Context modeling in few-shot molecular property prediction
>
> Thank you for showing this passage. This resolves my concerns about the context discussion.
>
> > FS-Mol experiment:
>
> - The performance values seem reasonable now.
> - The authors are able to show that their approach helps to boost PAR
> - I still think there are weaknesses in the FS-Mol experiment:
>     * PAR under-performs - compare performance reported in [6] with more suitable hyperparameters (differences are not that big though)
>     * All presented variants are outperformed by the Frequent Hitter model [6] which is a baseline which is not aware of any support set samples and simply learns average activity across tasks. Another backbone model - perhaps a GIN-encoder based ProtoNet or Neural Similarity Search version - might have been the better choice.
>
> Overall, I'd evaluate this paper to be a borderline paper now. Since I think the authors' manuscript has improved a lot and since the presented way of guiding fine-tuning by including molecular context is interesting, I'd adapt my score and slightly vote for acceptance.

---

> > ### Author Response · Authors · 2024-08-14
> > **Thank you!**
> >
> > Thank you so much for your response! We are very pleased that we have addressed your concerns. Your constructive and insightful comments have greatly helped us improve the quality of the paper. Once again, we sincerely thank you for your comments and participation in the discussion!
> >
> > Best regards,
> >
> > Authors

---

### Official Review · Reviewer_1DKF · 2024-07-12

**Soundness:** 3
**Presentation:** 2
**Contribution:** 2
**Rating:** 5
**Confidence:** 4

**Summary:**

This paper introduces a novel Pin-Tuning method. Focusing on improving the fine-tuning process of pre-trained molecular encoders, especially for the task of Few-Shot Molecular Property Prediction (FSMPP), Pin-Tuning skillfully balances the contradiction between the number of tunable parameters and the limited labeled molecular data, while reinforcing the encoder's context-awareness. The core of the approach is the introduction of MP-Adapter, a lightweight adapter for pre-trained message-passing layers, and Emb-BWC, a Bayesian weight consolidation scheme for pre-trained atom/key embedding layers. Experimental results show that Pin-Tuning exhibits excellent performance on public datasets, significantly improves the prediction performance in few-shot scenarios with fewer trainable parameters, and proves its effectiveness in the field of molecular attribute prediction.

**Strengths:**

1.The paper proposes Pin-Tuning methods, including MP-Adapter and Emb-BWC, which provide new solutions to the problem of fine-tuning pre-trained models in FSMPP tasks.
2.In the paper, it is proposed to integrate the context-aware capability into the adapter, which enhances the model's ability to perceive the molecular context and enables the model to be more effectively fine-tuned on few data samples.
3.The Pin-Tuning method proposed in the paper is evaluated on a public dataset, showing fewer trainable parameters and greater improvement in prediction performance, which demonstrates the effectiveness of the method.

**Weaknesses:**

1.The concept of Adapter has been widely studied and applied in the field of Natural Language Processing. In this paper, does MP-Adapter just take this concept over, or is there any design and optimization specific to molecular graphical neural networks (GNNs)? Is there any comparison with existing generalized frameworks for adapter networks?
2.The observation from the paper that the use of Identity matrix approximation works best in Emb-BWC seems to be a counterintuitive finding. This is because the Identity matrix approximation ignores possible correlations between parameters and assigns the same importance to each parameter. Is it true that correlations between parameters are less important in the molecular property prediction task than in other tasks? If so, does this reflect an intrinsic characteristic of molecular property prediction tasks?

**Questions:**

1.What does the a mean in Effect of weight of Emb-BWC regularizer λ in Sensitivity analysis? Is there a confusing typo here?
2.What are the advantages of the proposed MP-Adapter over existing adapter technology?
3.How to ensure the quality and reliability of the contextual information used in the paper?

**Limitations:**

Yes

---

> ### Author Rebuttal · Authors · 2024-08-07
>
> # Responses to Reviewer 1DKF
>
> We thank the reviewer for your constructive feedback. Please find detailed responses below.
>
> > `W1` `Q2`  Compared to adapters in NLP, the specific design and advantages of the proposed MP-Adapter.
>
> Based on your valuable feedback, we have provided a detailed description in the `Global Response` to clarify the specific design and considerations of our MP-Adapter for molecular representation fine-tuning. Additionally, we conducted experiments to compare the performance of fine-tuning molecular encoders using NLP adapters and our MP-Adapter, empirically demonstrating the advantages of our method. **We kindly suggest referring to our `Global Response` regarding this issue.**
>
> > `W2` Discussion on the experimental observations of Emb-BWC.
>
> Thank you for your insightful comments. This is a very meaningful question that deserves in-depth exploration. As we stated in Section 5.3, the results indicate that keeping pre-trained parameters to some extent can better utilize pre-trained knowledge, but the parameters worth keeping in fine-tuning and the important parameters in pre-training revealed by the Fisher information matrix are not completely consistent. Here, based on our experimental results, we provide a more in-depth discussion and our insights from the following two aspects:
>
> 1. `Importance of each parameter`: This involves two questions: `(a) Is it necessary to impose constraints on the importance of parameters?` `(b) What kind of importance assignment strategy is optimal for fine-tuning molecular representations?` For the first question, since the empirical results with three types of regularizers are better than those without any regularizer, it indicates that imposing importance constraints on the parameters is beneficial for fine-tuning molecular representations. For the second question, the observation is that the more relaxed the Emb-BWC constraint, the better the fine-tuning performance. The Emb-BWC constraint measures the importance of parameters in pre-training and uses this importance to constrain the fine-tuning process. **The explanation of this observation is that the important parameters in pre-training and the parameters that need to be retained during fine-tuning do not completely overlap, and some mechanisms of message passing require considerable updates during the fine-tuning process.**
> 2. `Correlation between different parameters`: Since the computation of the Hessian matrix in the original form of the Emb-BWC regularizer is intractable, we provide three diagonal approximation methods. Since all three approximation methods result in diagonal matrices, with non-zero values only on the main diagonal, they all assume that the contribution of each parameter update to the model performance is independent. In other words, **these three diagonal approximation methods imply that importance is assigned to each parameter independently, streamlining the correlations between parameters.** The off-diagonal values of the Hessian matrix can constrain the joint updates of parameters, but calculating these off-diagonal elements, whether through the original form or the approximated Fisher information matrix, is intractable due to the high-dimensional nature of the parameters. Therefore, **the better performance of the identity matrix approximation does not imply that the correlations between parameters are unimportant in molecular property prediction. Instead, it reflects that parameters which are not important during pre-training may have high importance during fine-tuning.**
>
> > `Q1` Is $a$ a typo for the weight $\lambda$ of the Emb-BWC regularizer in Section 5.4?
>
> Yes, the $a$ in the sensitivity analysis should be the regularization coefficient $\lambda$. This is a typo, and we greatly appreciate you pointing it out.
>
> > `Q3` How to ensure the quality and reliability of the context information?
>
> Since the molecular context is encoded based on the labels of the molecules in the target task and auxiliary tasks, **the accuracy and adequacy of these labels determine the quality and reliability of the contextual information**. When the required contextual labels are complete and accurately measured, the context is sufficiently reliable. However, the real situation is often not ideal, with some missing and noisy labels.
>
> **The GNN-based context encoder we adopted has already mitigated the issues of missing and noisy labels to some extent by propagating and smoothing information on the molecule-property bipartite graph**, thereby obtaining context representations that are robust to missing and noisy labels.
>
> To further improve the reliability of the context, **our potential solution is to rigorously measure and calibrate the uncertainty of the contextual labels**, such as measuring the entropy of the contextual label matrix. This could potentially further enhance the robustness of our method.

---

> ### Author Response · Authors · 2024-08-07
> **Thank you & Looking forward to your reply**
>
> Thank you very much for your precious time and valuable comments. We hope our responses have addressed your concerns. Please let us know if you have any further questions. We are happy to discuss them further. Thank you.
>
> Best regards,
>
> Authors

---

> > ### Comment · Reviewer_1DKF · 2024-08-13
> >
> > Thank you for your reply. Combined with the other comments and rebuttals, I maintain my initial rating.

---

> > > ### Author Response · Authors · 2024-08-14
> > > **Thank you!**
> > >
> > > Thank you very much for your response. If you have any further questions, feel free to ask and we would be more than delighted to answer.
> > >
> > > Best regards,
> > >
> > > Authors

---

### Author Rebuttal · Authors · 2024-08-07

# Global Response

We sincerely appreciate all the reviewers for your valuable feedback on our paper. In this global response, we aim to address the reviewers' concerns regarding the novelty and empirical advantages of our MP-Adapter. Specifically, we intend to answer the following questions:

>`The novelty of the MP-Adapter`: Considering that adapters have been widely used in natural language processing (NLP), what is the novelty of the MP-Adapter proposed in this paper for pre-trained message passing layers? In other words, what specific designs or considerations does the MP-Adapter have for molecular tasks and models?
>
>`The empirical advantage of the MP-Adapter`: Compared to using ordinary NLP adapters, what are the advantages in empirical performance when fine-tuning pre-trained molecular encoders with MP-Adapter?

The  insertion positions and structure of our proposed MP-Adapter are tailored to molecular representation learning, taking into account the architecture of molecular encoders and the need for perceiving molecular context. Specifically, the differences between our MP-Adapter and the ordinary NLP adapters widely used in NLP are as follows:

1. **The position where the MP-Adapter is inserted into the pre-trained molecular encoder is determined by considering the overall architecture of the adapted molecular encoder and the target module.** In NLP, the adapted model are typically pre-trained transformer-based large language models. Whether it is an encoder-only, encoder-decoder, or the currently popular decoder-only language model backbone, the basic units are stacked transformer layers, which are the target for adaptation by NLP adapters. NLP adapters are usually inserted after the `multi-head attention` modules or `feed-forward` modules to fine-tune them parameter-efficiently, as these modules conduct the most critical operations in the transformer layers [1,2]. In molecular representation learning, the pre-trained molecular encoders that need to be adapted use graph neural networks (GNNs) as their model backbone. Molecular encoders follow the message passing mechanism, comprising `atom/bond embedding layers`, `message passing layers consisting of aggregation function and update function`, and the final `readout function`. The insertion positions of our MP-Adapter are specifically designed for the message passing mechanism of the GNNs. Since the update function in the message passing layer is the source of GNNs' high expressive power [3] and has the most complex parameters, the MP-Adapter is inserted after the update functions to adapt them.
2. **Our MP-Adapter takes both the output of the message passing layers and the encoded molecular context as inputs, thereby enabling fine-tuning under the guidance of the molecular context.** Unlike the adapters in NLP, our MP-Adapter not only achieves parameter-efficient fine-tuning of the pre-trained parameters but also incorporates the encoded molecular context as an additional input to the adapter, achieving in-context tuning. In Section 4.2, based on the significance of molecular context, we propose the method for encoding molecular context and incorporating it into the MP-Adapter. The initial Eq. 3 is updated to Eq. 7, reflecting the consideration of molecular context, which is also a special design specifically for fine-tuning molecular encoders.

We compared the impact of MP-Adapter and NLP Adapter on fine-tuning pre-trained molecular encoders, and the results are presented in the table below. The three models being compared are not equipped with the proposed Emb-BWC to ensure that the only difference lies in the choice of adapter. In the table, `No Adapter` represents the most state-of-the-art baseline method GS-Meta, `NLP Adapter` adds a vanilla NLP adapter after the message passing layers in GS-Meta, and `MP-Adapter` equips the message passing layers with our context-aware MP-Adapter, without our proposed Emb-EWC constraint. For each experiment, we run 10 times with different seeds and report the average ROC-AUC score.

|                                   | Tox21   |        | SIDER   |        | MUV     |        | PCBA    |        |
| --------------------------------- | ------- | ------ | ------- | ------ | ------- | ------ | ------- | ------ |
|                                   | 10-shot | 5-shot | 10-shot | 5-shot | 10-shot | 5-shot | 10-shot | 5-shot |
| No Adapter                        | 86.67   | 86.43  | 84.36   | 84.57  | 66.08   | 64.50  | 79.40   | 77.47  |
| NLP Adapter                       | 87.94   | 87.80  | 85.18   | 84.79  | 71.86   | 69.58  | 79.75   | 78.35  |
| MP-Adapter                        | 90.17   | 89.59  | 92.06   | 91.43  | 72.37   | 71.65  | 80.74   | 78.51  |
| Pin-Tuning (MP-Adapter + Emb-BWC) | 91.56   | 90.95  | 93.41   | 92.02  | 73.33   | 70.71  | 81.26   | 79.23  |

From the results above, it can be observed that using the vanilla NLP Adapter can improve the fine-tuning performance in few-shot scenarios, thanks to the reduction in the number of tunable parameters and our decision on the appropriate insertion positions. Our MP-Adapter further introduces molecular context information, which is crucial for molecular property prediction, into the fine-tuning process, resulting in better performance.

**References**

[1] LLM-Adapters: An Adapter Family for Parameter-Efficient Fine-Tuning of Large Language Models. EMNLP 2023

[2] Efficient Large Language Models: A Survey. TMLR 2024

[3] How powerful are graph neural networks? ICLR 2019

---

### Decision · Program_Chairs · 2024-09-25

**Decision:**

Accept (poster)

**Comment:**

The paper introduces Pin-Tuning, a parameter-efficient fine-tuning approach specifically designed for GNN-based few-shot molecular property prediction tasks. This approach features a novel adapter layer (MP-Adapter) and a Bayesian weight consolidation scheme (Emb-BWC), both of which are combined to demonstrate superior performance across benchmark datasets. The authors' thorough rebuttal, which included additional experiments and clarifications, addressed the reviewers' concerns, highlighting the method's relevance and potential impact. This work offers a novel approach that will positively contribute to few-shot molecular property prediction literature and will be of significant interest to the NeurIPS audience.